# Nuclear oligo hashing improves differential analysis of single-cell RNA-seq

Hyeon-Jin Kim [1,5], Greg Booth [1,5], Lauren Saunders [1], Sanjay Srivatsan[1], José L. McFaline-Figueroa [2] & Cole Trapnell [1,3,4 ✉]

Single-cell RNA sequencing (scRNA-seq) offers a high-resolution molecular view into complex tissues, but suffers from high levels of technical noise which frustrates efforts to compare the gene expression programs of different cell types. "Spike-in" RNA standards help control for technical variation in scRNA-seq, but using them with recently developed, ultra-scalable scRNA-seq methods based on combinatorial indexing is not feasible. Here, we describe a simple and cost-effective method for normalizing transcript counts and subtracting technical variability that improves differential expression analysis in scRNA-seq. The method affixes a ladder of synthetic single-stranded DNA oligos to each cell that appears in its RNA-seq library. With improved normalization we explore chemical perturbations with broad or highly specific effects on gene regulation, including RNA pol II elongation, histone deacetylation, and activation of the glucocorticoid receptor. Our methods reveal that inhibiting histone deacetylation prevents cells from executing their canonical program of changes following glucocorticoid stimulation.

[1] Department of Genome Sciences, University of Washington, Seattle, WA 98195, USA. [2] Department of Biomedical Engineering, Columbia University, New York City, NY 10027, USA. [3] Brotman Baty Institute of Precision Medicine, Seattle, WA 98195, USA. [4] Allen Discovery Center for Cell Lineage Tracing, Seattle, WA 98195, USA. [5] These authors contributed equally: Hyeon-Jin Kim, Greg Booth. ✉email: coletrap@uw.edu

Single-cell RNA sequencing (scRNA-seq) methods have revolutionized our understanding of development[1–5] and disease[6–12]. Comparing transcript counts for one or more genes between populations of single-cells is a fundamental step in nearly all such experiments. However, current single-cell RNA-seq protocols exhibit high levels of technical noise and variability[13]. Current scRNA-seq methods detect RNA molecules by first reverse-transcribing them into cDNA, and inefficiencies in this conversion results in a failure to count the overwhelming majority of RNA molecules. Moreover, RNA content varies widely across individual cells even of the same type (e.g., as a function of cytoplasmic volume)[14]. Therefore, it is often difficult to assess whether an observed difference in the abundance of a gene transcript between cells is due to technical or biological variability. Moreover, inherent cell-to-cell variation in gene expression may obscure biologically meaningful differences to begin with. Thus, removing technical variation in transcript counts through proper normalization across cells could greatly improve the sensitivity of single-cell RNA-seq studies.

In order to normalize transcript abundances across cells, most scRNA-seq analyses compute library "size factors" that scale each cell's counts relative to the others so that cells with few total read counts are not ignored in favor of cells in which more molecules were counted[15,16]. Such methods assume that all cells have similar total RNA content and the variation in their levels is purely technical. By assuming all cells in an experiment have equal total transcript abundances, size factor normalization precludes detection of global differences in transcript production and limits differential analysis to changes in relative abundance of individual transcripts. To combat the limitations of size factor normalization, a second approach compares transcript counts against an external standard. This approach works by "spiking in" several species of synthetic RNA molecules into the lysate of each cell at a range of concentrations[17]. These synthetic transcripts are detected along with endogenous mRNAs and by comparing the observed synthetic RNAs to their concentrations, one can estimate the efficiency of RNA detection in each cell, improving downstream analyses[18]. Importantly, the use of an external standard enables one to detect changes in global transcription[19].

Using external spike-in controls with widely used scRNA-seq platforms is impractical or cost-prohibitive for several reasons. Droplet-based scRNA-seq instruments such as the 10X Chromium require most droplets to be "empty" in order to ensure that non-empty droplets contain only one cell. Adding a fixed amount of external RNA to each droplet would necessitate sequencing through the synthetic RNA in the acellular drops, vastly increasing the cost of the experiment[20]. Current single-cell RNA-seq methods based on combinatorial indexing such as sci-RNA-seq[1] are also largely incompatible with the use of spike-in RNAs. Introducing synthetic RNAs to each cell in a controlled manner would require physically isolating or manipulating it, eliminating the key scalability advantage combinatorial indexing enjoys over other methods.

To enable the use of spike-in controls in a cost-effective manner, we developed an external molecular standard compatible with combinatorial indexing-based scRNA-seq methods. In sci-RNA-seq, permeabilized, fixed cells are first split across the wells of a 96- or 384-well microtiter plate. Reverse transcription is then performed within the intact cells or nuclei in situ using primers that carry barcode sequences corresponding to each well, yielding cDNA that is indexed according to the well in which it was transcribed. Cells are then pooled and split into a new plate and the cDNAs are indexed again by PCR. The resulting RNA-seq library fragments can be sequenced as a pool and deconvolved by collecting sets of reads that have the same pair of well-identifying indexes, yielding molecular profiles of thousands of individual

cells without requiring their physical isolation. Improved versions of the protocol introduce additional rounds of splitting, indexing, and pooling to collect transcriptomes of millions of cells in one experiment[2]. Our molecular standard is affixed within each cell and processed as if they are endogenous mRNA molecules, which enables normalization of transcript counts from single cells and offers better control of technical variation in large scale scRNA-seq experiments.

To develop a method for introducing external normalization standards into single cells, we exploited nuclear oligo hashing, a way of irreversibly labeling cells with synthetic DNA barcodes. We previously introduced nuclear oligo hashing as part of sci-Plex, in which polyadenylated single-stranded oligonucleotides ("hashes") are used to label nuclei and enable multiplexed single-cell transcriptome profiling of millions of cells from thousands of conditions in one experiment[21]. In a sci-Plex experiment, cells from each condition are permeabilized and incubated with hash oligos unique to that condition, which become trapped within the nuclei and can be chemically fixed in place. The oligos subsequently serve as templates for reverse transcription reactions and are therefore captured alongside endogenous mRNA transcripts during sci-RNA-seq library preparation.

We reasoned that, in addition to labeling cells according to condition, hashes could be added to cells at varying and known concentrations and then used as an external proxy for estimating and subtracting technical noise between individual cells. Assuming that hash oligos become trapped in the nuclei in a concentration-dependent manner, we designed a ladder of distinct hash oligos, with each hash species present at a unique, predetermined concentration within the mixture (Fig. 1a). After incubating exposed nuclei with the ladder, we then fix and collect the nuclei and subject them to sci-RNA-seq. To normalize transcriptome counts from each nucleus, we compare the observed hash ladder counts to the concentrations for each species, constructing a calibration curve for each cell by fitting a negative binomial regression model. Using the estimated parameters of this regression, we compute a cell-specific normalization constant, which should account for technical variations in sci-RNA-seq data. We can also use these regressions to identify and discard low-quality cells.

## Results

**Hash ladders can be used as spike-in controls in sci-RNA-seq experiments.** As a proof of concept, we designed a ladder comprising eight different hash oligos, theoretical abundance ranging from 0.1–12.8 picomoles per one million nuclei, and introduced into a sci-RNA-seq library preparation of HEK293T cells during the cell lysis step (Fig. 1a). As expected, we recovered unique molecular identifier (UMI) counts from both the endogenous mRNA molecules and the individual hash oligos from the hash ladder, with about 12% of the sequencing used for the hash ladder (Supplementary Fig. 1a, b). The observed number of hash oligo UMI counts globally reflected the expected abundance of each hash oligos in the ladder, and the hash ladder counts between each cell in this experiment were well correlated (median coefficient of determination between cells: 0.966) (Fig. 1b and Supplementary Fig. 1c). For individual cells, we constructed a calibration curve describing the relationship between the expected and observed number of counts of each hash molecule by fitting a negative binomial regression model (Fig. 1c).

We hypothesized that the goodness of fit of the hash ladder calibration curve reflects the extent of technical variability introduced in the library preparation steps and therefore could be used as a quality control to identify low quality cells. To test

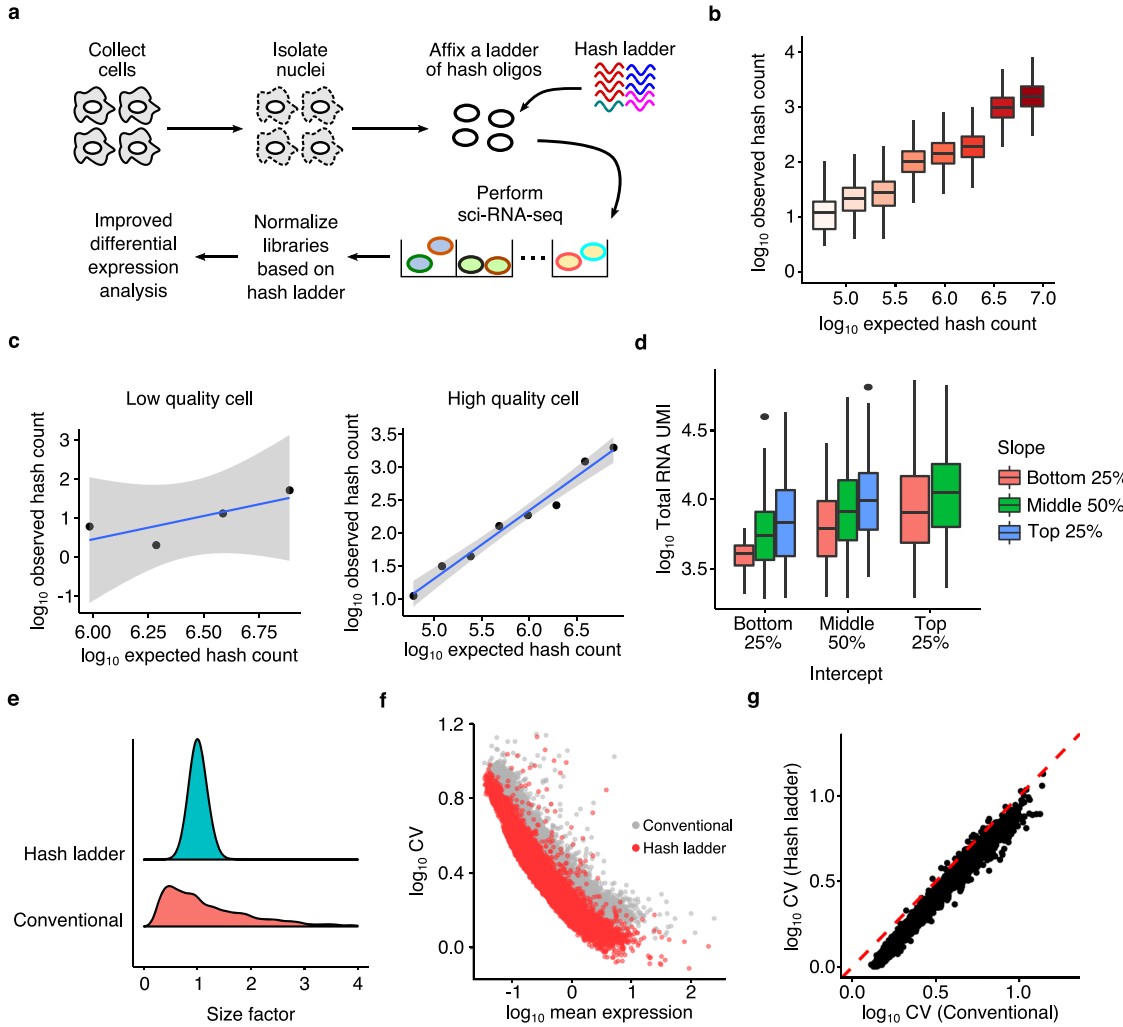

**Fig. 1 A set of hash oligos can be captured within nuclei and serve as external standards in sci-RNA-seq experiments. a** An experimental overview of the hash ladder method. Nuclei are isolated from cells, fixed with a ladder of hash oligos, then processed with sci-RNA-seq. **b** Boxplot of hash oligo UMI counts per cell, each hash oligo spiked in at different abundance ($n = 1884$ cells). **c** Scatter plot of expected and observed hash ladder UMI counts, demonstrating a cell with low (left) and high (right) hash capture efficiency. **d** Boxplot of total RNA count of cells grouped according to the intercept and slope of their hash ladder calibration line ($n = 1884$ cells). **e** Distribution of cell-specific size factors computed using the conventional and hash ladder methods. **f** Scatter plot of normalized average expression and coefficient of variation (CV) of expressed genes in HEK293T. **g** Comparison of CV values computed with the conventional and hash ladder-based normalization methods. The centerline of the boxplots in **b** and **d** indicates the median, the box displays the first and third quartile, and the whiskers show the 1.5 interquartile range (IQR). Outliers are displayed as points.

this hypothesis, we examined the relationship between the quality of the fit of each cells' regression model for its hash counts and its overall endogenous RNA UMIs. Cells with high recovery of endogenous RNA molecules tended to have hash regression models with higher pseudo R-squared values than those with low endogenous RNA UMI counts (Supplementary Fig. 1d). In subsequent analyses, we considered cells to be low-quality if the pseudo R-squared value of the fit was lower than 0.7 or total hash ladder UMI count was less than 100. After filtering out low quality cells, we obtained transcriptomes (median 7787 UMIs) and hash ladder calibration curves (median 3108 UMIs) for 1884 cells (99.6%), with median pseudo R-squared value of 0.954 (Supplementary Fig. 1e).

We next evaluated whether the slope and intercept values of the calibration line reflect the technical variability in the total abundance of the endogenous mRNA molecules. Grün et al. proposed that the slope of external, spike-in based calibration curves captures the library preparation efficiency whereas the intercept value captures the fraction of endogenous and spike-in

RNA used for library preparation and recovered by sequencing[18]. In this experiment, hash ladders were spiked into a homogeneous population of HEK293T cells; therefore, we hypothesized that the technical variation in the total RNA abundance should be captured by the hash ladder parameters. Indeed, the total RNA abundance was positively correlated with the values of slope and intercept when they were binned according to their values, indicating that the observed variation in the hash ladder counts can be used to estimate the technical variation across transcript counts within a cell (Fig. 1d). To examine whether our external spike-in would be applicable for more complex and hetero-geneous biological samples, we applied our oligo hashing method to isolated nuclei from dissociated whole zebrafish embryos. We found that recovery of hash molecules from the zebrafish nuclei did not vary significantly across different cell types, suggesting that hash uptake and retention is not cell type dependent (Supplementary Fig. 2). These results demonstrate the potential utility of our hash ladder spike-in approach to various biological models.

Next, we aimed to develop a procedure to normalize each cell's transcriptome profile based on the hash ladder. Conventional normalization approaches for scRNA-seq data scale a cell's gene expression values by a size factor proportional to the cell's total recovered RNA count[15,16]. These approaches reduce cell-specific technical bias by converting the raw expression data to relative measurements under the assumption that the majority of genes are not differentially expressed between the cells[15,16]. However, they fail in cases where changes in global expression are expected across different biological conditions[22]. To address this issue, we formulated an approach to compute global cell-specific size factors using the hash ladder parameters (see Methods). The hash ladder-based size factors are computed using the total hash ladder UMI count, slope and intercept of the calibration curve to correct for cell-specific technical bias that arise from library preparation and sequencing. Unlike conventional size factor normalization, the hash ladder-based size factor normalization corrects for cell-specific technical biases that arise from library preparation and sequencing without transforming the gene expression data to relative abundances. Size factors derived from the hash ladder were symmetric about 1, while conventional size factors exhibited a long right tail corresponding to cells with excess total RNA counts (Fig. 1e). Compared with conventional normalization, when cells were normalized using their individual hash ladder-based size factors, the coefficient of variations (CV) were lower for almost all genes with the hash ladder-based approach (Fig. 1f, g).

**Hash ladders facilitate analysis of changes to global transcription in single cells**. We next evaluated whether hash ladder-based normalization could be used to accurately detect global changes in mRNA transcript levels. Flavopiridol inhibits the activity of a transcription elongation factor P-TEFb, thereby repressing mRNA transcription across the genome[23,24]. We treated HEK293T cells with 300 nM of flavopiridol for increasing amounts of time followed by sci-RNA-seq with a ladder composed of 48 different hash oligos. We obtained gene expression profiles and hash ladder calibration models for 1370 high-quality single cells (Fig. 2a and Supplementary Fig. 3). Consistent with inhibition of transcription elongation, cells exposed to flavopiridol for the longest times showed the greatest reduction in RNA recovery per-cell (Fig. 2b), and after 24 h of flavopiridol treatment, we recovered 55.90% fewer total RNA UMIs per cell on average.

We then compared the effects of conventional and hash ladder-based normalization approaches on pseudotime ordering and differential expression analyses. First, we used Monocle3[2] to visualize the data using Uniform Manifold Alignment and Projection (UMAP)[25] and order cells along the flavopiridol pseudotime trajectory. The UMAP projections obtained with the conventional and hash ladder normalization approaches were qualitatively comparable, and both trajectories were consistent with actual treatment times (Fig. 2c and Supplementary Fig. 4a). We used Monocle3 to test for changes in each gene's expression as a function of exposure to flavopiridol. When transcript counts were normalized with conventional size factors, Monocle3 reported an equal number of significantly upregulated and downregulated genes, despite the well characterized global repression of RNA Pol II transcription elongation induced by flavopiridol[26–28] (Fig. 2d). Importantly, such artifacts are characteristic library size-based normalization[29]. By contrast, hash ladder-based normalization yielded 406 more downregulated genes and 31 fewer upregulated genes (Supplementary Fig. 4b, c and Supplementary Table 1). As an example, flavopiridol is known to block the expression of genes involved in cell adhesion[30,31]. Unlike the results from conventional

normalization, the hash ladder normalization captured a consistent decreasing expression of genes known to be involved in endothelial cell adhesion (*CD151* and *LAMC2*) across flavopiridol treatment time (Fig. 2e and Supplementary Fig. 4d). Ultimately, the magnitude of log2 fold changes (24 h vs. vehicle) computed with hash ladder normalized expression values was on average higher for downregulated genes and lower for upregulated genes compared to that of conventional normalized expression values, suggesting a general improvement in sensitivity to changes in transcript abundance in this system (Fig. 2f and Supplementary Fig. 4e). The observed effect sizes of differentially expressed genes were further corroborated when we repeated the FP time course with bulk RNA-seq measurements normalized with ERCC spike-ins (Supplementary Fig. 5). Altogether, these results demonstrate the value of unbiased external normalization for detecting global changes in transcription, enabled through the use of nuclear hash ladders in single cell RNA-seq experiments.

**Histone deacetylase inhibition transiently reduces global transcript levels**. Next, we sought to use hash ladders to investigate the role of histone deacetylases (HDAC), an important class of chemotherapeutics, in regulating transcription. Histone deacetylases remove acetyl groups from histones and other proteins and are thought to act as transcription repressors. Inhibition of HDACs leads to cell-cycle arrest and hyperacetylation of histones, altering the relative expression levels of many genes[32], but the mechanisms by which these genes are regulated has not been fully characterized. In principle, hyperacetylation of histones could facilitate access of transcriptional machinery to many genes and broadly increase transcription across the genome. Chromatin also may also serve as a reservoir of acetate, and acetate flux through chromatin could help the cell buffer against changes in available acetate or pH[33]. We recently demonstrated that HDAC inhibition (HDACi) deprives cells of acetyl-coenzyme A (acetyl-CoA), and cells compensate by activating alternative pathways for the biosynthesis or import of acetyl-CoA precursors (e.g., citrate) in dose-dependent manner[21]. Individual cells exhibited dramatically heterogeneous responses to HDACi. For example, even at doses that kill a substantial fraction of cells, we captured many that were transcriptionally indistinguishable from vehicle-treated controls. Therefore, gene expression changes in response to HDACi (e.g., activation of tumor suppressors) could be a consequence of a cellular "metabolic crisis" characterized by acetyl-CoA deprivation. As acetyl-CoA is important for many cellular processes including transcription, HDACi might also lead to a reduction in global transcription. We therefore sought to disentangle the relative contributions of changes to chromatin structure and cellular metabolism to gene regulation in response to HDAC inhibition.

To assess the effects of HDACi on global transcript levels in single-cells, we first performed a time-series HDAC inhibition experiment to estimate the time point at which acetyl-CoA deprivation is first detectable. We treated A549 cells with one of two HDAC inhibitors (abexinostat or pracinostat) for 0, 0.5, 1, 3, 6, 12, or 24 h and performed sci-Plex with the hash ladder, obtaining transcriptomes and hash ladder calibration lines for 1548 cells ($n = 2$ replicates, Supplementary Fig. 6). As with the flavopiridol time-course experiment, both normalization approaches ordered cells according to their treatment time (Fig. 3a). When viewed with conventional normalization, total mRNA counts remained stable over time. In contrast, the hash ladder-based normalization revealed a dramatic but transient reduction of total RNA levels along the HDAC inhibitor pseudotime trajectory (Fig. 3b). Over this trajectory, we detected an early 60.7% reduction in mRNA levels in cells at the nadir, but after 24 h of treatment with HDACi, cells had fully restored

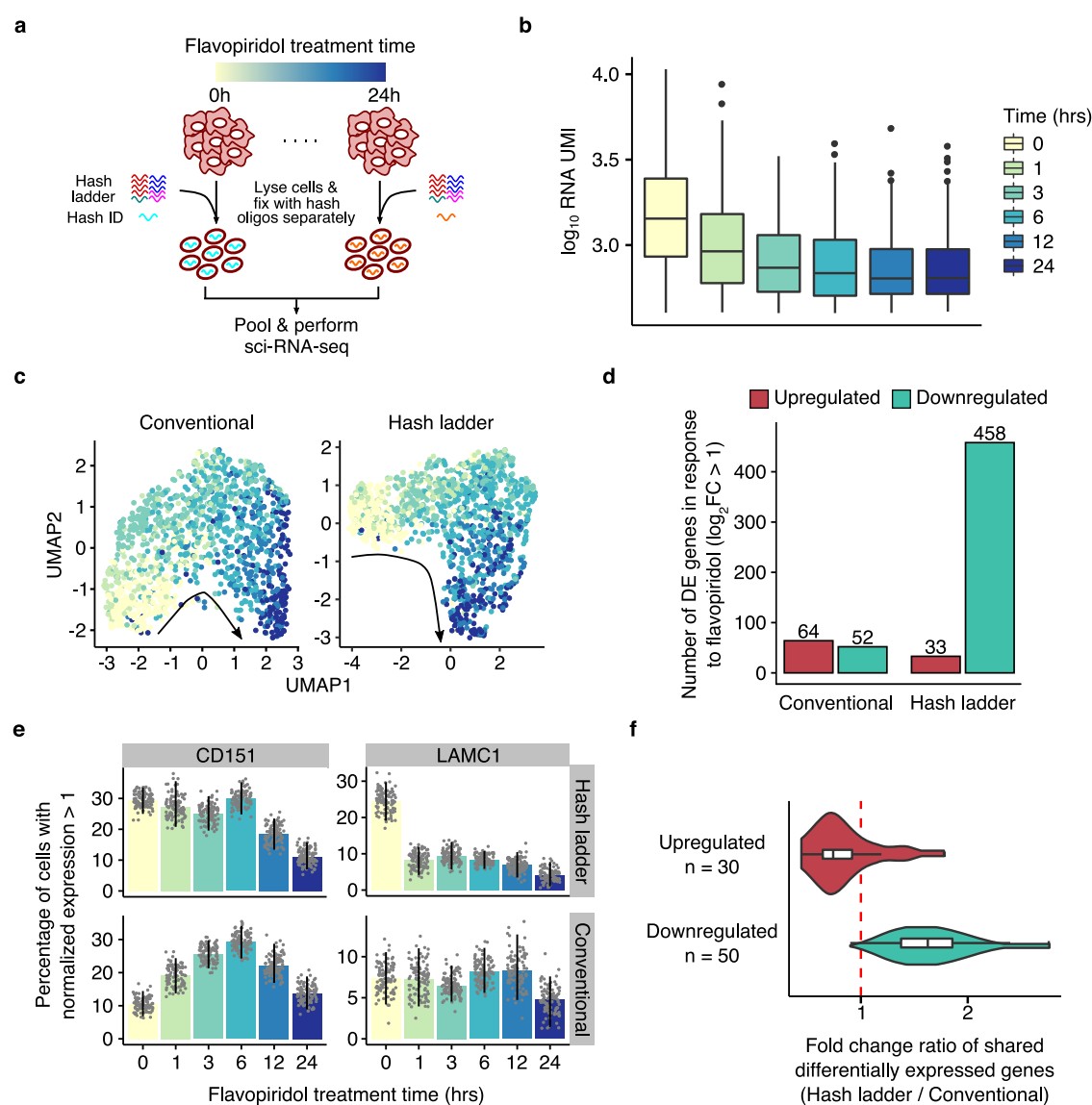

**Fig. 2 Hash ladder expands our ability to detect global reduction in transcript levels caused by flavopiridol. a** Overview of the experiment. HEK293T cells were treated with flavopiridol for different periods of time and labeled with a ladder of hash oligos and additional hash oligo for multiplexing prior to sci-RNA-seq preparation. **b** Boxplot showing total RNA UMI counts for cells treated with flavopiridol at different time points ($n = 1370$ cells). **c** UMAP projections of flavopiridol treated HEK293T cells colored by treatment time and normalized by conventional (left) and hash ladder (right) size factors. **d** Barplot showing number of differentially expressed genes in response to flavopiridol using the conventional and hash ladder normalization approaches. **e** Conventional and hash ladder normalized expression levels of *CD151* and *LAMC1* at different flavopiridol treatment times. Bars represent the percentage of cells with normalized expression value greater than 1, and the error bars show the 95% confidence interval obtained using a bootstrap method ($n = 100$ bootstrap samples). **f** Violin plot showing the ratio of effect size estimates of common differentially expressed genes computed with hash ladder vs. conventional normalization. The centerline of the boxplots in **b** and **f** indicates the median, the box displays the first and third quartile, and the whiskers show the 1.5 IQR. Outlier values are displayed as points.

mRNA levels. Interestingly, the transient reduction in mRNA levels was not detected when the single-cell data were aggregated by discrete treatment times or in bulk RNA-seq data, likely owing to and consistent with the large heterogeneity in cellular response to HDAC inhibition we previously observed[21,34] (Supplementary Fig. 7).

Nevertheless, differential expression analysis revealed that cells at later timepoints had undergone dramatic changes in gene expression (Supplementary Table 2). Hierarchical clustering of 446 differentially expressed genes revealed four kinetically distinct groups (Fig. 3c). Gene set enrichment analysis[35] using Gene Ontology[36] and MSigDB hallmark[37] gene groups showed enrichment of genes involved in cell cycle, cellular metabolism,

and immune response, consistent with previous findings[21,38,39] (Supplementary Table 3). This analysis suggests that despite returning to normal mRNA levels, HDAC-inhibited cells shift from a proliferative gene expression program to one that helps compensate for acetyl-CoA deprivation. Compared to the conventional approach, hash ladder normalization recovered a greater number of differentially expressed genes previously identified from a larger, published sci-Plex experiment, including upregulated genes involved in acetyl-CoA biosynthesis (Supplementary Fig. 8). Moreover, in line with established perturbations of proliferation by HDACi[40,41], we observed a group of HDACi-treated cells with altered gene expression relating to cell cycle effects (Supplementary Fig. 9).

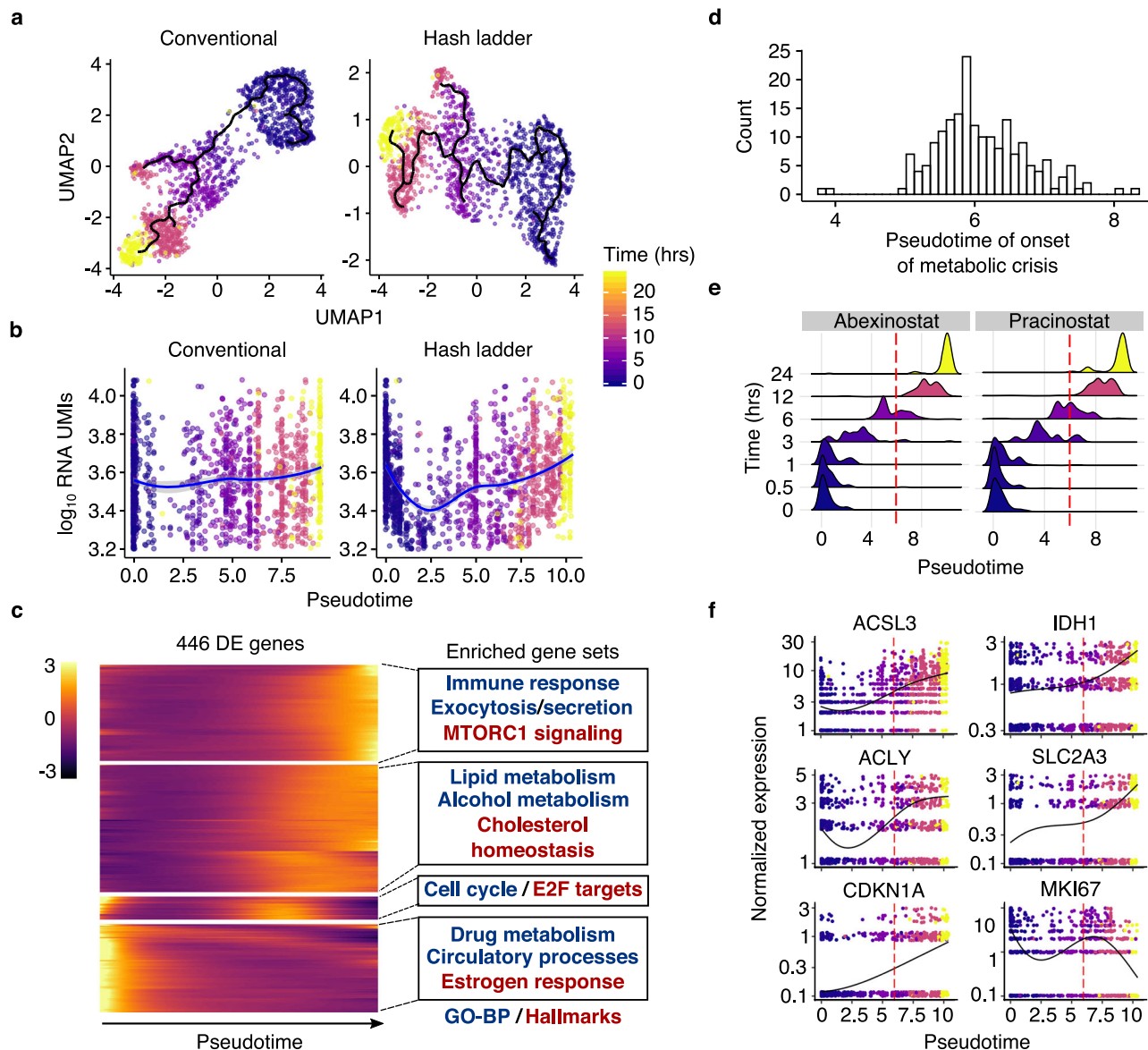

**Fig. 3 Histone deacetylase (HDAC) inhibitor time course trajectory pinpoints onset of acetate starvation. a** UMAP projections of HDAC inhibitor (abexinostat and pracinostat) treated A549 cells (n = 1548 cells over two replicates) colored by treatment time and normalized by conventional (left) and hash ladder (right) size factors. The HDAC inhibitor trajectories are overlaid onto the UMAP projections. **b** Scatterplot showing total RNA UMIs as a function of pseudotime position obtained from the conventional (left) and hash ladder (right) size factor normalized data. **c** Hierarchical clustering of 446 of 1485 highly differentially expressed genes along the HDAC inhibitor pseudotime trajectory (likelihood ratio test, FDR < 1 × 10$^{-10}$ and number of expressed cells > 100). Rows represent row centered and z-scaled dynamics of gene expression. **d** Histogram of pseudotime points at which the centered, z-scaled expression value is equal to zero for upregulated genes involved in cellular metabolism (n = 188 genes). **e** Pseudotime distribution of HDAC inhibitor treated cells at each treatment time point. The red dotted line represents the median pseudotime of distributions from Fig. 3c. **f** Hash ladder normalized expression of genes involved in cellular metabolism and cell cycle across pseudotime, marked by onset of acetate starvation (dotted red line).

We next attempted to disentangle the direct impact of HDACi on chromatin acetylation and transcription from the secondary effects on the transcriptome arising from acetyl-CoA deprivation. Reasoning that there might be a window of time where HDACi prevented histone deacetylation but acetyl-CoA was not yet depleted, we sought to identify the time at which metabolic-driven effects of HDAC inhibition set in. To estimate the onset of acetyl-CoA deprivation, we assessed the expression patterns of 188 upregulated genes involved in cellular metabolism. Using the HDAC inhibitor pseudotime trajectories from Monocle3, we defined the pseudotime at which each gene rose appreciably above its baseline untreated level (Fig. 3d and

Supplementary Fig. 8d). The median pseudotime across all 188 genes corresponded to a position of the trajectory populated by cells harvested at approximately six hours following exposure to HDAC inhibitor (Fig. 3e). Our estimated onset of acetyl-CoA deprivation was consistent with the expression patterns of genes that compensate for acetyl-CoA deprivation, including those involved in acetyl-CoA biosynthesis (*ACLY*, *ACSL3*), citrate metabolism (*IDH1*), and glucose uptake (*SLC2A3*) (Fig. 3f). Importantly, the nadir of cellular total mRNA levels preceded acetyl-CoA deprivation, suggesting that reduction in transcript levels is not simply a consequence of the "metabolic crisis" that arises from inhibiting histone deacetylases.

**Short-term histone deacetylase inhibition attenuates the glucocorticoid response to dexamethasone**. We next sought to investigate whether HDACi prevented cells from transcriptionally responding to external stimuli. Glucocorticoid receptor (GR), a transcription factor that modulates diverse biological processes such as stress and inflammatory responses, directly interacts with histone deacetylases[42]. We examined the ability of cells to mount a transcriptional response to synthetic glucocorticoid agonist dexamethasone (DEX), both before and after onset of acetyl-CoA deprivation induced by HDAC inhibitors. We treated A549 cells with one of two HDAC inhibitors (abexinostat or pracinostat) for either 4 or 24 h prior to DEX treatment and performed sci-Plex with a hash ladder included for normalization ($n = 2$ replicates, Fig. 4a and Supplementary Figs. 10 and 11). Since we observed that upregulation of enzymes used to compensate for acetyl-CoA deprivation occur after 6 h of HDAC inhibitor treatment, we reasoned that inhibiting HDACs for only 4 h would allow us to distinguish direct effects of histone hyperacetylation from secondary metabolic effects on the cell. In addition to vehicle-treated controls for each condition, we performed the experiment using two doses (1 µM or 10 µM) of each HDAC inhibitor to consider dose-dependent effects. After filtering out low-quality cells, we obtained gene expression profiles (median 3816 UMIs/cell) and hash ladder calibration lines (median 260 UMIs/cell) for 3710 high-quality cells.

Consistent with distinct transcriptional effects before and after metabolic effects of HDAC inhibitor treatment, clustering of hash ladder normalized sci-RNA-seq profiles was mainly driven by the duration of HDAC inhibitor treatment, rather than by HDAC inhibitor dose (Fig. 4b). In the absence of HDAC inhibitor, vehicle and DEX treated cells formed separate clusters in the UMAP embedding, with DEX treatment increasing the expression of GR activated genes, including *ANGPTL4, FKBP5*, and *TSC22D3* (Supplementary Fig. 12). By contrast, cells that received HDAC inhibitor prior to DEX treatment intermixed with those that only received the HDAC inhibitor treatment, suggesting they failed to mount a strong DEX-induced glucocorticoid response (Fig. 4b). Moreover, the loss of a distinguishable response to dexamethasone was seen after just 4 h of HDACi treatment, suggesting that failure to mount a GR response is not solely due to acetyl-CoA deprivation.

We next performed differential gene expression analysis (Supplementary Tables 4 and 5) amongst the various treatment groups. Of the 171 DEX-responsive genes we identified by comparing DEX-treated to untreated cells, only 67 (39%) and 57 (33%) genes properly responded to DEX in cells which had been treated with an HDAC inhibitor for 4 and 24 h, respectively (Fig. 4c). These genes included those involved in hypoxia response and tumor necrosis factor alpha (TNF-A) signaling pathways (Supplementary Fig. 13a, b). The DEX response appeared primarily impacted by the duration of HDAC inhibition rather than dose, as we observed no genes with significant dose-dependent changes in transcriptional outcome between cells pretreated with 1 µM and 10 µM of HDACi.

We then assessed whether the magnitude of transcriptional changes induced by DEX is impacted by HDAC inhibition. Indeed, the DEX-induced transcriptional changes were attenuated by prior HDAC inhibition for 93% of the genes, suggesting a severely compromised activation of GR response by DEX in HDAC inhibitor treated cells (Fig. 4d). Interestingly, the attenuation of the DEX response was more severe after the 4 h HDAC inhibitor treatment than after 24 h of exposure (Supplementary Fig. 13c, d). For example, the GR-induced activation of *TSC22D3*, a glucocorticoid-induced transcriptional regulator of anti-inflammation[43], was significantly impaired after 4 h HDAC inhibitor treatment (log2 fold change of 1.65), but only slightly impaired after 24 h HDAC inhibitor treatment (log2 fold change of 3.43). In the case of *TSC22D3*, the recovery of GR-inducibility after extended HDACi resembles its behavior during inflammatory signaling, when activation of *TSC22D3* coincides with prolonged hyperacetylation of histones[44–46] and metabolic reprogramming[47].

HDACi-treated cells failed to regulate a majority of GR response genes, even after only 4 h of exposure to inhibitors, suggesting that disrupting histone acetylation dynamics is sufficient to interfere with the glucocorticoid response (Fig. 4e). Broadly, these genes were enriched for roles in managing reactive oxygen species, which are known to be affected by HDAC inhibition[48–50] (Supplementary Fig. 14). Unresponsive genes also included G-protein coupled receptors associated with cytoskeleton reorganization, concordant with previous reports of HDAC inhibition preventing glucocorticoid-induced hypertension[51] and changes in microtubule dynamics[52]. Interestingly, we observed various classes of effects on these unresponsive genes suggesting multiple ways in which HDACi might interfere with the DEX response (Supplementary Fig. 15). GR-regulated genes that fail to respond after 4 h treatment with HDACi but do respond at 24 h included several members of the complement system (Fig. 4f). This is consistent with the effects of HDAC inhibition in regulating innate immune pathways[38,53,54]. In contrast, cells treated for only 4 h successfully regulated genes involved in fatty acid metabolism, while cells treated for 24 h did not, suggesting that acetyl-CoA deprivation may interfere with their regulation. Importantly, fatty acid metabolism can contribute to acetyl-CoA synthesis[55], raising the possibility that the cells' compensation for acetyl-CoA deprivation may override their response to DEX.

## Discussion

Here, we show how a ladder of single-stranded DNA molecules can be incorporated into sci-RNA-seq experiments as an external normalization control, revealing changes in global transcript levels and the expression of individual genes. By affixing a set of hash oligos at predetermined concentrations to each nucleus, we demonstrate that a calibration curve can be constructed for individual cells, controlling for cell-to-cell technical variation introduced during library preparation. As a demonstration of the utility of hash ladders in single cell transcriptomics, we were able to dramatically enhance the detection of global repression in transcription caused by cyclin-dependent kinase inhibitor flavopiridol.

We then applied the hash ladder normalization approach to reveal dynamic changes in global transcriptional output and pinpoint the onset of acetate deprivation caused by histone deacetylase treatment. We anticipated that if anything, hyper-acetylated chromatin in HDAC-inhibited cells would facilitate transcription, leading to increased expression of many genes. Surprisingly, our analysis showed that inhibiting histone deacetylases transiently leads to a global reduction in mRNA output and interferes with the cell's ability to mount a transcriptional response to dexamethasone treatment. Both effects of HDACi treatment preceded the onset of acetyl-CoA deprivation, suggesting that they are not simply a consequence of metabolic changes in the cell. Taken together, these results lend support to the notion that chromatin serves as a reservoir of acetate for the cell, and that acetate flux through chromatin is important for cell metabolism and potentially transcription of many genes[21,34].

Our analysis suggests that hash ladder-based normalization is particularly beneficial when changes in global transcriptional levels are expected (e.g., flavopiridol). By contrast, when only a small number of genes were affected by a treatment, conventional normalization performed similarly to the hash ladder-based

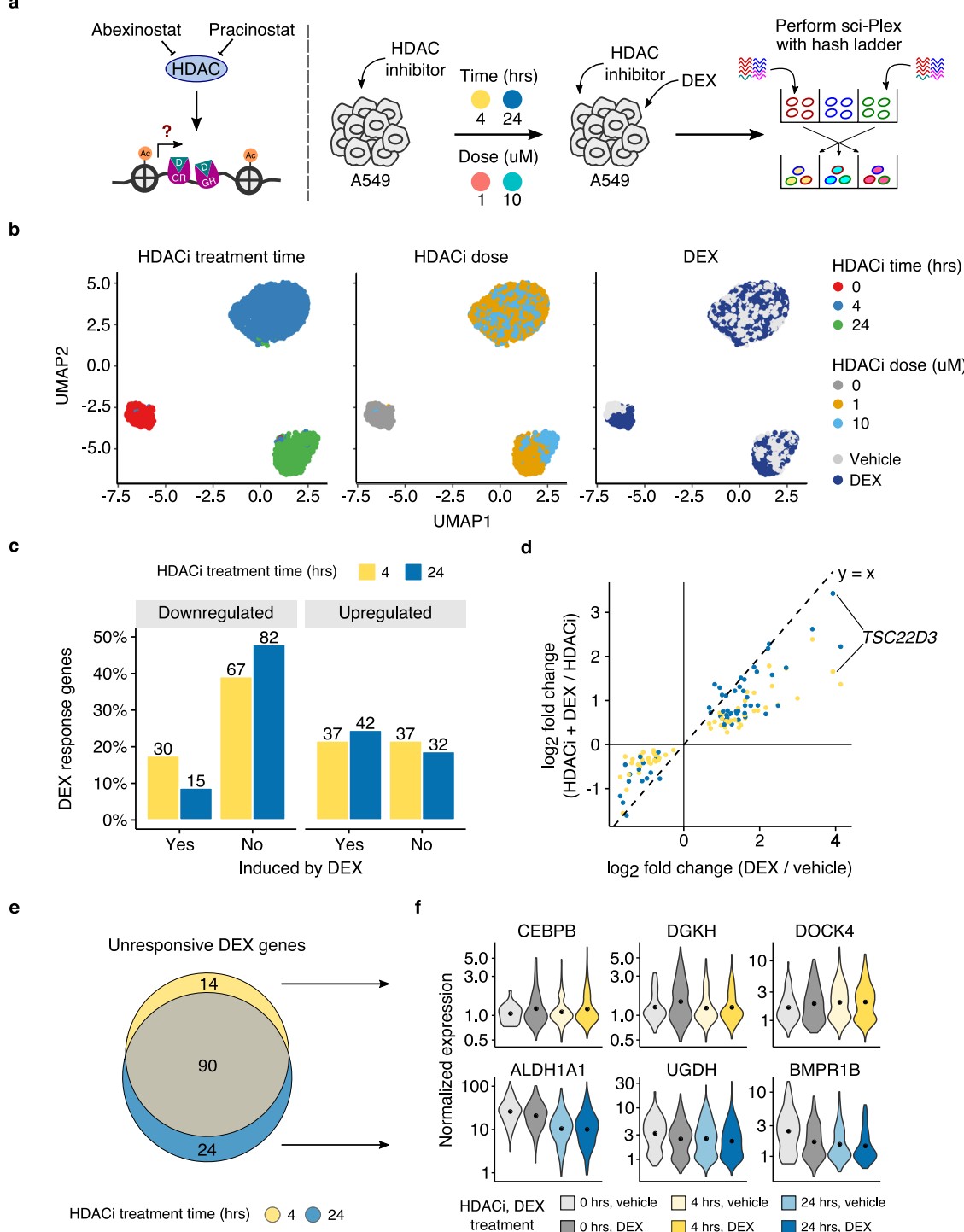

**Fig. 4 HDAC inhibition prevents mounting of dexamethasone (DEX) response. a** Overview of the experiment. A549 cells were treated with HDAC inhibitors for different periods of time and dose prior to two hour DEX treatment. These cells were subjected to sci-Plex with the addition of the hash ladder. **b** UMAP projections of vehicle- and DEX- treated A549 cells ($n = 3710$ cells over two replicates) in the absence and presence of HDAC inhibitors using the hash ladder normalized expression values. **c** Percentage of DEX response genes that do not respond to DEX treatment with prior HDAC inhibitor treatment. **d** Scatter plot comparing log2 fold changes of DEX responsive genes (DEX/vehicle) without and with preceding HDAC inhibitor treatment. Colors respond to the duration of HDAC inhibitor treatment. **e** Venn diagram of genes that do not respond to DEX at different HDAC inhibitor treatment times. **f** Hash ladder normalized expression of DEX genes that do not respond in cells that received 4 h (top) and 24 h (bottom) preceding HDAC inhibitor treatment (bottom).

normalization (e.g., HDACi and HDACi/DEX). However, because it is often difficult or impossible to predict the outcome of previously unseen perturbations, the hash ladder provides a robust normalization approach, unbiased by a priori assumptions.

Hash oligos are a versatile tool for single-cell transcriptome sequencing. As demonstrated in our HDAC inhibitor time course and dexamethasone experiments, hash oligos can easily be used both for sample multiplexing and as external standards in single-cell combinatorial indexing-based methods. This approach therefore enables gene expression profiling of many experimental conditions, including those with heterogeneous starting populations, with improved normalization. The dynamic range of a hash ladder can be easily tuned to assess the limits of detection and be used to model noise structure in scRNA-seq data[13,18]. Furthermore, hash oligos can be captured using aldehyde and alcohol fixatives (MeOH; unpublished result); therefore, hashing could be implemented in other scRNA-seq platforms that are compatible with chemical fixation.

Importantly, the hash ladder spike-in approach assumes that uptake of hash oligos is uniform across all nuclei in the experiment, and it is important to consider situations where this might be violated, such as with differences in cell cycle stage, cell size and/or cell or nuclear permeability. While it is challenging to correlate permeability and hash uptake, we believe that the effect of cell cycle/size on hash ladder normalization is minimal and found that an observed difference in hash uptake did not significantly alter differential expression analysis between these cells (Supplementary Figs. 16 and 17).

In summary, the use of a hash ladder offers an unbiased and versatile normalization tool that is simple, low-cost, and compatible with the highly scalable single-cell combinatorial indexing RNA sequencing method.

## Methods

**Cell culture**. A549 (CCL-185 from ATCC) and HEK293T (CRL-3216 from ATCC) cells were cultured in DMEM (Gibco) media containing 10% fetal bovine serum (Invitrogen) and 1% penicillin and streptomycin (Gibco) at 37 °C with 5% $CO_2$.

For the flavopiridol time course experiment, HEK293T cells were seeded onto a 6-well culture plate at a density of $4 \times 10^5$ cells per well in 2 mL of media. For the HDAC inhibitor time course and HDAC inhibitor and dexamethasone co-treatment experiments, A549 cells were seeded onto a 96-well culture plate at a density of $2.5 \times 10^4$ cells per well in 100 μL of media.

**Drug treatment**. Cells were grown for 24 h after they were seeded onto the cell culture plates. For the flavopiridol time course experiment, 0.9 μL of 1 mM flavopiridol (Selleck Chemicals) was added to each 6 well to attain a final concentration of 300 nM. For the HDAC inhibitor time course experiment, 1 μL of 1 mM of either abexinostat (Selleck Chemicals) and pracinostat (Selleck Chemicals) was added to each 96 well to attain a final concentration of 10 μM. Similarly, for the HDAC inhibitor and dexamethasone co-treatment experiment, we added 1 μL of 100 μM or 1 μL of 1 mM of either abexinostat or pracinostat to each 96 well to attain a final concentration of 1 and 10 μM, respectively. Dexamethasone (Selleck Chemicals) was added at a concentration of 100 nM (1 μL of 10 μM) two hours before the end of the HDAC inhibitor treatment. DMSO (10%) was used as a vehicle for flavopiridol and HDAC inhibitor treatments and ethanol (10%) was used as a vehicle for dexamethasone treatment. The HDACi timecourse and HDACi/DEX co-treatment experiment was performed in duplicates, and the flavopiridol timecourse experiment was only performed once.

**Design of hash ladder**. The structure of hash oligos is previously described by Srivatsan et al.[21]. The capture of hash ladder by nuclei is determined by factors including, but not limited to, sample processing, total RNA content, and sequencing depth. For mammalian cell lines, we estimate that the nuclear capture rate of hash oligos ranges from 0.1–1%. We empirically determined that the ladder should be constructed so that approximately 6–8 million hash molecules are spiked in for each cell or nucleus (assuming that each cell/nucleus takes up equal amounts of hash molecules in the solution) to obtain a median hash ladder UMI count of 1000–5000. For the pilot experiment, we used a hash ladder consisting of 8 different hash oligos, covering from 0.1–12.8 picomoles per one million nuclei. For the rest of our experiments, we spiked in a hash ladder consisting of 48 different hash oligos, ranging from 0.25 attomoles −20 femtomoles for 100,000 nuclei (flavopiridol) and for each 96 well (HDACi and dexamethasone). The detailed

information on how the hash ladder mixture can be made is provided in Supplementary Table 6.

**Cell harvest, nuclei isolation, and hash oligo capture**. To harvest the cells, media was removed and cells were washed with DPBS (Gibco) and dissociated off the plate using trypLE (Gibco). Trypsinization from trypLE was quenched with an equal volume of ice cold FBS containing DMEM media. Cells were pelleted by centrifugation at $500 \times g$ for 5 min, washed with ice cold DPBS, and resuspended in ice cold DPBS. For the initial proof of concept experiment and flavopiridol time course experiments, cells were then counted with a hemocytometer using 0.4% Trypan Blue. Approximately 100,000 cells were pelleted at $500 \times g$ for 5 min and resuspended in 1 mL of ice cold lysis buffer (10 mM Tris-HCl, pH 7.4, 10 mM NaCl, 3 mM $MgCl_2$, 0.1% IGEPAL CA-630) supplemented with 1% Superase RNase Inhibitor (Invitrogen) and 4.8 μL of the hash ladder mixture. After lysis with gentle pipetting, cells were fixed by addition of 4 mL of fixation buffer (5% paraformaldehyde in 1.25X PBS) on ice for 20 min. After the cells were fixed, they were washed with 1 mL of nuclei suspension buffer (10 mM Tris-HCl, pH 7.4, 10 mM NaCl, 3 mM $MgCl_2$, 1% Superase RNase Inhibitor, 1% 0.2 mg/mL NEB-Next BSA) and resuspended in 100 μL of nuclei suspension buffer (NSB).

For the HDAC inhibitor time course and HDAC inhibitor and dexamethasone co-treatment experiments, cells were pelleted at $500 \times g$ for 5 min and transferred to a 96-well V-bottom plate containing 100 μL of ice cold lysis buffer supplemented with 1% Superase RNase Inhibitor, 400 femtomoles of hash ID oligos and 4.8 μL of the hash ladder mixture in each well. Cells were then lysed with gentle pipetting, and 200 μL of fixation solution was added to each well and incubated on ice for 20 min. After fixation, the nuclei were pooled into a trough and transferred into a 50 mL conical tube. Nuclei were then pelleted by centrifugation at $500 \times g$ for 5 min and washed with 1 mL of NSB twice before resuspending in 100 μL of NSB.

**Preparation of sci-RNA-seq libraries**. The sci-RNA-seq libraries were prepared following the protocols as previously described by Cao et al.[2] and Srivatsan et al.[21]. Isolated and hashed nuclei were permeabilized in 500 μL of permeabilization buffer (0.25% Triton-X in NSB) for 3 min on ice and then washed once with 1 mL of NSB. After the wash step, nuclei were pelleted at $500 \times g$ for 5 min, and resuspended in 100 μL of NSB. The nuclei were then counted using a hemocytometer, and the concentration of the nuclei was adjusted to 2.5 million cells per mL. To each well of skirted twin.tec 96 well LoBind plate (Fisher Scientific, cat no. 0030129512), a mixture of 5000 nuclei, 0.25 μL of 10 mM dNTP mix (Thermo Fisher Scientific, cat no. R0193), and 1 μL of 25 μM uniquely indexed oligo-dT[1] was added, followed by denaturation at 55 °C for 5 min and immediate reannealing on ice. Next, 1.75 μL of reverse transcription mix (1 μL of Superscript IV first-strand buffer, 0.25 μL of 100 mM DTT, 0.25 μL of Superscript IV (Thermo-Fisher) and 0.25 μL of RNA-seOUT (Invitrogen) recombinant ribonuclease inhibitor) was added to every well incubated at 55 °C for 10 min and placed on ice. Reverse transcription was stopped by adding 5 μL of stop solution (40 mM EDTA and 1 mM spermidine) to each well. Nuclei were then pooled using wide bore tips for transfer to a trough, stained with DAPI at a final concentration of 3 μM, and finally transferred to 5 mL flow cytometry tube (Falcon) through the 0.35 μm filter cap. FACS Aria II cell sorter (BD) was used to sort 25–50 nuclei per well into 96 well LoBind plates containing 5 μL of Buffer EB (Qiagen). Nuclei were gated based on DAPI-A vs DAPI-H to ensure sorting of single nuclei.

After sorting, second strand synthesis was performed by adding 0.75 μL of second strand mix (0.5 μL of mRNA second strand synthesis buffer and 0.25 μL of mRNA second strand synthesis enzyme, New England Biolabs) to each well and incubating at 16 °C for 180 min. Tagmentation was performed by adding 6 μL of tagmentation mix (0.02 μL of a custom TDE1 enzyme in 6 μL 2× Nextera TD buffer, Illumina) to each well and incubating for 5 min at 55 °C. The tagmentation reaction was stopped by adding 12 μL of DNA binding buffer (Zymo) and incubating for 5 min at room temperature. Each well was then purified by using 36 μL (1.5X) of Ampure XP beads (Beckman Coulter), eluted in 16 μL of EB buffer and transferred to a new 96 well LoBind plate.

Each well was mixed with 2 μL of 10 μM indexed P5 primer, 2 μL of 10 μM indexed P7 primer[1] and 20 μL of NEBNext High-Fidelity master mix (New England Biolabs) and PCR was carried out using the following program: 72 °C for 3 min, 98 °C for 30 s and 19 cycles of 98 °C for 10 s, 66 °C for 30 s and 72 °C for 1 min followed by a final extension at 72 °C for 5 min. The PCR products were then all pooled, concentrated using a DNA clean and concentrator kit (Zymo) and purified with 0.8X Ampure XP beads. Library concentrations were measured by Qubit (Invitrogen) and fragment sizes were visualized using TapeStation HS D1000 DNA Screen tape (Agilent). Libraries were sequenced on a Nextseq 500 platform (Illumina) using a high output 75 cycle kit (Read 1: 18 cycles, Read 2: 52 cycles, Index 1: 10 cycles and Index 2: 10 cycles).

**Pre-processing of sci-RNA-seq data**. The sci-RNA-seq libraries were processed according to the protocol described in ref. [21]. First, Illumina reads were demultiplexed using bcl2fastq (v2.20). Custom scripts were used to extract barcodes that match reverse transcription (RT) indices within Levenstein distance cutoff of 2. After RT barcode matching, poly(A) sequences were trimmed using trim-galore, and the trimmed reads were aligned to *hg38* using STAR aligner (v2.5.2b) with

default settings. The aligned reads were filtered with MAPQ ≥ 30, deduplicated, and collapsed by unique molecular identifiers (UMIs). The deduplicated reads were then assigned to genes using bedtools intersect function with an annotated human gene model. Knee plots were then generated from UMI counts per cell barcode to filter out true cell barcodes from debris. Thresholds were selected on a per-experiment basis and gene expression profiles from cell barcodes with total UMI counts greater than the knee threshold were used to generate a CDS object for downstream analysis.

**Sample assignment with ID hash oligos**. The hash IDs were assigned sample labels as described in ref. [21]. In summary, reads from hash oligos were extracted from the demultiplexed read files by matching the first 10 bp of read 2 to the hash barcodes with the Levenstein distance of 2 and looking for trailing poly(A) sequences from position 12–16 bp. These reads were deduplicated and collapsed by UMIs, which were then used for sample assignment and hash ladder calibration line construction.

Nuclei with hash UMIs that did not significantly vary from a background hash UMI distribution were filtered out (FDR < 0.01). The background distribution was estimated for each experiment by averaging the hash UMIs from cell indices that are likely from debris fragments. The debris cells were obtained by filtering for cell indices with fewer than an empirically determined RNA UMI cutoff. To assign sample labels and filter out doublets, enrichment ratios were calculated for each cell by taking the ratio of the most abundant vs. the second most abundant hash ID. As described in ref. [21], the enrichment ratio cutoff is determined empirically and carefully chosen to separate unlabeled and singly labeled nuclei. For the flavopiridol and HDAC inhibitor time course and DEX co-treatment experiments, we used the enrichment cutoff of 10.

**Construction of hash ladder calibration curves**. Hash counts from the ladder were considered separately from the hashes which label a cell's sample (described above). After extracting hash read counts for each nucleus, nuclei were required to have a minimum number of unique ladder molecules recovered, as well as a total hash ladder UMIs > 100 for downstream analysis. For the experiments performed with the ladder of 48 hash oligos, nuclei with fewer than 10 unique hash ladder molecules were discarded. The observed read counts for each hash oligo in the hash ladder were then compared against the expected number of reads to construct a hash ladder calibration curve for each cell. The expected or theoretical abundance of each hash oligo was estimated by dividing the number of molecules in the ladder by the number of starting cells in each condition. A negative binomial regression model was then fit to each calibration line to obtain the slope and intercept values. To isolate cells with high quality hash ladder calibration lines, nuclei with pseudo R-squared value of less than 0.7 were removed.

**Computation of cell-specific hash ladder-based size factor**. To normalize for cell-specific bias in scRNA-seq data, gene expression values are often divided by its cell-specific size factor. Conventionally, cell-specific size factors are computed by taking the geometric mean of the log total RNA UMIs from all cells in the experiment[15,16]. Therefore, conventional size factors are proportional to the cell's total RNA UMIs:

$$f_i \sim x_i \qquad (1)$$

where $f_i$ is conventional size factor value and $x_i$ is total RNA UMI counts for cell $i$.

Here, we propose an alternative method to compute cell-specific size factors using the hash ladder parameters:

$$f_i{'} \sim \log\left(\frac{z_i}{h_i}\right) * \frac{m_i}{-b_i} * r_i \qquad (2)$$

where $f_i{'}$ is the size factor derived from the hash ladder, $z_i$ is total hash UMI count, $h_i$ is the duplication rate for hash oligos, $m_i$ is the slope of the hash ladder calibration line, $b_i$ is the intercept, and $r_i$ is the RNA duplication rate for cell $i$. In our analysis, the slope and intercept values are derived from fitting a negative binomial regression line to the expected and observed hash ladder counts.

With the assumption that the nuclear capture rate of the hash oligos are approximately equal across cells, we reasoned that technical variation should mostly account for the differences in the total number of observed hash UMIs. The distribution of total hash UMIs were log-normally distributed and therefore logarithm of total hash UMIs used for the calculation. Additionally, we reasoned that the slope and intercept values of the hash ladder calibration curves reflect the cells' library preparation efficiency and the amount of input RNA/hash ladder molecules used for the library and sequencing[18], respectively. For example, variations in reverse transcription (RT) reaction efficiency will influence the capture of lowly expressed RNA molecules (slope) and systematic errors in pipetting and pooling steps will affect the number of molecules captured in each reaction (intercept). The duplication rates of the hash oligos and transcriptomes were included in the size factor calculation to correct for differences in sequencing depth between samples and multiple sequencing runs. As a result, nuclei that exhibit high technical variation as indicated by the hash ladder parameters would have low size-factor values to compensate for low library preparation efficiency.

**Dimensionality reduction and pseudotime ordering**. We analyzed our sci-RNA-seq data using Monocle3[2]. For dimensionality reduction, gene expression profiles were first divided by the cell-specific size factors (conventional or hash ladder) and log transformed after adding a pseudocount of 1. The normalized gene expression profiles were then reduced to 50 principal components with principal component analysis (PCA) and then they were further reduced to two-dimensional Uniform Manifold Approximation and Projection (UMAP) space using the reduce_dimension function in Monocle3.

For the flavopiridol and HDAC inhibitor time course experiments, cells were clustered with the Louvain community detection method[56]. As described by Cao et al.[2], the cells ordered along a drug treatment pseudotime trajectory using the functions learn_graph, and order_cells in Monocle3. The exact details of the function parameters used for each experiment can be found in the provided code.

**Differential expression analysis**. To perform differential expression analysis, we used the fit_models function in Monocle3. For each experiment, we performed differential expression analysis twice, using each normalization method.

For the flavopiridol and HDAC inhibitor time course experiments, we fit the normalized gene expression profiles using the model:

$$\log(y_i) \sim t' \qquad (3)$$

where $y_i$ is the negative binomial variable that captures the UMI count of gene $i$ and $t'$ represents pseudotime values that are smoothed via a natural spline function with 3 degrees of freedom to capture the dynamic expression patterns along the pseudotime trajectory.

For the HDAC inhibitor and dexamethasone co-treatment experiment, we extracted DEX-induced genes using the non-HDAC inhibitor treated cells with the model:

$$\log(y_i) \sim d \qquad (4)$$

where $d$ is a binary variable that indicates whether a cell has received the dexamethasone treatment. A gene is interpreted as DEX responsive if it has a significant DEX term (FDR < 0.05). To evaluate whether the transcriptional changes induced by DEX are prevented by differential cellular effect of HDAC inhibition, we performed differential expression analysis on 4 and 24 h HDAC inhibitor treated cells separately, using the same model formula as above with the additional terms that reflect the dose and drug information of the HDAC inhibitors.

To determine whether the HDAC inhibitor treatments alone alter the expression of the DEX response genes, we separately analyzed the cells that have received the HDAC inhibitor at each timepoint (4 and 24 h) using the model formula:

$$\log(y_i) \sim d \qquad (5)$$

Additionally, we included terms such as PCR plate identity and HDAC inhibitor drug information, to the model to further regress out technical batch effects.

**Visualization and clustering of HDAC inhibitor pseudotime dependent genes**. To visualize and cluster the differentially expressed genes that vary along the HDAC inhibitor pseudotime trajectory, we first identified DE genes with FDR < 1e −10 and number of expressed cells >100. The smoothed pseudotime fitted gene expression values were centered and z-scaled, and the resulting gene expression x pseudotime matrix was visualized and clustered with the ward.D2 method using the pheatmap package in R.

**Gene set enrichment analysis**. We used the runGSAhyper function from the R *piano* package to perform gene set enrichment analysis on a list of differentially expressed (DE) genes. Briefly, runGSAhyper function uses Fisher's exact test to evaluate whether the DE genes are enriched for particular gene sets against the background of expressed genes in the experiment. For our analysis, we used the Gene Ontology biological processes and MSigDB hallmark gene set collections, and we used the false discovery rate threshold of 0.05 for calling enriched gene sets.

**Determination of onset of acetyl-CoA deprivation**. To determine the onset of acetyl-CoA driven cellular metabolic crisis due to histone deacetylase inhibition, we assessed the expression pattern of upregulated genes that are involved in metabolic processes such as glucose, alcohol, and fatty acid metabolism. Specifically, we filtered for the upregulated HDAC inhibitor pseudotime dependent genes using the metabolic process Gene Ontology terms. We then centered and z-scaled the pseudotime fitted expression patterns of these genes. We defined the time point at which the centered, z-scaled expression value exceeds zero as the onset of acetyl-CoA metabolic crisis. We determined this time point for all the upregulated metabolic genes and used the median time point as a global estimate. We followed this procedure for both the hash ladder normalized and conventional size factor normalized expression values.

**Annotation of unresponsive DEX genes in presence of HDAC inhibitors**. By performing differential expression analyses, we identified a set of normally DEX-

induced genes that become *unresponsive* to Dex as a consequence of HDAC inhibition. We also identified a set of normally DEX-induced genes that are altered due to HDAC inhibitor treatment. With these sets of genes, we annotated the *unresponsive* DEX genes as either saturated, dominated, or attenuated. Saturated indicates the gene is differentially expressed as a consequence of HDAC inhibition and in the same direction as the response to DEX alone. Similarly, a gene was defined as dominated (meaning expression changes were dominated by the HDAC inhibitor) if the gene is differentially expressed as a consequence of HDAC inhibition but in the opposite direction as DEX alone. Finally, an *unresponsive* DEX gene was classified as attenuated if the expression is not significantly altered by HDAC inhibition alone, but nonetheless incapable of responding to DEX after HDACi.

**Reporting summary**. Further information on research design is available in the Nature Research Reporting Summary linked to this article.

## Data availability

Raw and processed data used in this study have been deposited on GEO under accession number GSE166470 and available on our GitHub repository (see Code availability).

## Code availability

The scripts used for this manuscript are available on Zenodo (https://doi.org/10.5281/zenodo.6374386) or GitHub at https://github.com/khj3017/hash_ladder.

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

## Acknowledgements

We thank all members of the Trapnell lab for helpful discussions and feedback. This work was funded by grants from the NIH (R01HG010632, U54HL145611, and RC2DK114777) and the Paul G. Allen Frontiers Group.

## Author contributions

C.T. initiated and supervised the project. H.J.K., G.B., and C.T. designed experiments. H.J.K. and G.B. performed experiments. H.J.K. analyzed the data. L.S. and S.S. performed the experiment with zebrafish. J.L.M.F. provided technical support. C.T., H.J.K., and G.B. wrote the manuscript with the support of other authors.

## Competing interests

C.T. is a SAB member, consultant, and/or co-founder of Algen Biotechnologies, Altius Therapeutics, and Scale Biosciences. One or more embodiments of one or more patents and patent applications filed by the University of Washington may encompass methods, reagents, and the data disclosed in this manuscript. Some work in this study is related to technology described in patent applications. All other authors declare no competing interests.
