## [Peer Review File · Nature Communications]

Reviewers' Comments:

Reviewer #1:

Remarks to the Author:

In this manuscript, the authors describe the application of nuclear oligo hashing to improve quantification of single-cell RNA seq analysis. This method relies on spiking in a ladder of synthetic single strand DNA oligos to single cell nuclei preparations. These oligos get attached to the nuclei and then are co-processed in the subsequent single-cell sequencing workflow. This hashing method has previously been developed by the authors to enable barcoding for large scale single cell transcriptome studies after chemical perturbation. Here the authors use a ladder of hash oligos at different concentrations and develop computational methods to apply them for improved normalization of scRNA-seq data. The authors first show that the new normalization method improves CVs from a HEK293 scRNA-seq experiment. In a second application, they show that in contrast to normalization by conventional size factors, their method enables the detection of a global transcriptional downregulation with flavopiridol. Finally, they use HDAC inhibition in A549 cells as a model for temporal changes. They observe an initial global transcriptional downregulation, that seems compensated only later through the upregulation of acetyl-CoA biosynthetic pathways. Finally, they show that HDAC inhibition prevents the response to the glucocorticoid dexamethasone particularly at an early time point

In summary, this manuscript provides a technical advance and novel biological data on HDAC inhibition that will be of interest to the readers of Nature Communications after the following points have been addressed:

1, Particularly for quantitative applications, it appears important to mention potential other biases that the new method might introduce. For example, uptake of the hash oligos might not be perfectly uniform across all nuclei and might correlate with other factors including nuclear size or cell cycle stage. The authors should ideally provide experimental data on single cell distribution of hash oligo uptake in the cell lines they use (e.g. using fluorescently labeled hash oligos with a FACS readout) or at least mention such potential biases in their manuscript.

2, For the HDAC inhibitor treatment, the authors describe a 60% drop in overall mRNA levels in their early 1 and 3 hour timepoints. How would this effect be compatible with average mRNA half-lives in mammalian cells? It is important to validate this effect in an independent method, either on bulk cells or using a nascent transcription method.

3, For the dexamethasone treatment, the UMAP representations in Fig. 4B might be misleading as the non-HDAC inhibitor treatment condition is shown separately whereas the two time points of HDAC inhibitor treatment are shown together, whereby treatment time is clearly the main driver of separation. Either all experiments (no HDAC inhibitor, 4 h, 24 h treatment) should be shown in one plot, or separate plots should be shown for all doses/timepoints always comparing vehicle vs. dexamethasone as done for the non-HDAC inhibitor treated condition. The chosen HDAC inhibitor concentrations appear too high to elicit a differential effect.

4, From Fig. 4F it appears that at least some of the genes that are unresponsive to dexamethasone treatment in the presence of HDAC inhibitors are unresponsive because HDAC inhibition already causes a similar effect to dexamethasone treatment. Eg. DOCK4 appears upregulated by HDAC inhibition alone already, and no further upregulation occurs with dexamethasone. Such effects cannot be recognized from the representation chosen in Fig. 4D, and it would be desirable to include an additional heatmap.

5, I am not fully clear how the treatment regimen for HDAC inhibitors and dexamethasone was performed. I think it was 4 hours HDAC inhibitor treatment followed by 2 hours of HDAC inhibitor plus dexamethasone, rather than two plus two hours. In this case the relevant reference is 6 h HDAC inhibitor treatment, which may be too long given that the authors already detect compensatory mechanisms at this time point.

Reviewer #2:

Remarks to the Author:

The manuscript describes the potential to utilize nuclear hash oligo as a 'spike-in' control for RNA quantification from single-nuclei RNA-seq. By deriving the size factor from the fitted slope and intercept of hashing oligo ladders, and applying them to the situation where the global transcriptional changes are expected, authors have shown nicely that nuclear hash oligo based normalization results in increased power for the detection of differentially expressed genes.

This manuscript address very important topic in single-cell analysis and elegantly conducted overall. I have few minor comments to improve the manuscript for the publication.

(1) Justifying the logics for the size factor calculation.

Could the authors test their size factor calculations, by testing the inclusion/exclusion of each factors and showing whether it changes the conclusion (in terms of differential gene expression testing)?

(2) Extrapolating the size factor normalization to the case where you don't have spike-in

In most experiments, there are still no spike-in included. Could authors investigate whether the true size factor calculated based on spike-in can be derived/explained from the values that can be obtained in more general single-cell RNA-seq techniques (such as total RNA UMIs, number of genes detected)? This can be done by simply showing correlation plots or fitting regression models. If the authors could suggest some general rule that can improve the current 'crude' size factor calculation, that it can have a greater impact on the field.

(3) Could the authors discuss about the usability of this hashing methods? i.e. could this be applied in other platform including 10X Genomics, and what will be the general success rate of adding right concentration of spike-in. (as having it too much could potentially ruin your experiment)

(4) (optional) cell cycle systems (such as the one with FUCCI reporter) could be perfect one to show the power of this approach

Jong-Eun Park

Reviewer #3:

Remarks to the Author:

Kim et al., present a novel spike-in "hashing" strategy that normalizes scRNA-seq data to correct for technical variations without having to assume that total mRNA levels are similar between individual cells, an important point when comparing conditions that affect global transcript levels. The authors demonstrate that the hashing strategy detects global transcript changes (i.e., after treatment with inhibitors), while conventional normalization methods fail to capture these global changes. However, the hashing and conventional strategies produce comparative results when small, more specific transcript changes occur (Figure 4 and Supp Fig.9).

Given the lack of spike-in normalization methods currently available for scRNA-seq data and the gain in popularity of scRNA-seq experiments, this study is timely and important for the field. The study is well designed and presented in its current form, but the manuscript would benefit from clarification of a few points and minor additions to the analysis prior to publication.

1. In Supplementary Figure 3D only ~25-28 unique hash oligos were recovered per cell out of the 48 used in the hash ladder. Is there a bias in the oligos that are not recovered across cells or do different oligos fail to sequence across different cells? If there is a bias in the oligos that are not recovered, then is it necessary to include them in the ladder? If random oligos fail to sequence across individual cells, do differences in which oligos are recovered affect normalization?

2. In Figure 3C, 446 genes were used for the hierarchical clustering that revealed four kinetically distinct groups enriched in various gene ontologies, yet in Supp Figure 6A there are 1,485 differentially expressed genes identified by hash normalization. What is the rationale for only using

446 genes for the clustering and gene ontology analysis in Figure 3C?

3. Which ontology terms are enriched for the 643 genes that were identified by both the conventional and hash normalization methods after HDACi treatment (Supp Figure 6A)? Do most of the genes involved in acetyl-CoA biosynthesis fall into the 643 genes commonly identified by the two methods or are they mostly found in the gene set only identified by the hashing strategy?

4. Could the authors please elaborate on the following statement and figure? "Additionally, the UMAP embedding of gene expression profiles using the hash ladder-based normalization approach displayed a more pronounced branching of cells undergoing cell cycle related changes as a result of HDAC inhibition, which is less well defined in the embedding obtained with conventional normalization (Fig. 3A and Supplementary Fig.7)."

What is the significance of a more pronounced branching of cells? This point may not be very clear for readers that are not as familiar with this type of analysis.

5. The majority of detected differentially expressed DEX genes are identified by both conventional and hash normalization, with ~20 genes that show opposite responses (Supp Fig 9A and B). Could the authors please clarify if the majority of expected DEX response genes were detected using both normalization methods or are most DEX responsive genes present in the ~20 genes that showed opposite responses between the two normalization methods?

6. Do the authors have any insight into why ~20 genes displayed opposite responses between the two normalization methods after DEX treatment (Supp Fig 9A and B)? Is there something about the expression of these genes that make them more susceptible to differences between the normalization methods?

7. If the majority of expected DEX response genes are identified by both normalization methods, does this suggest that the main benefit of the hash normalization method is to normalize transcript counts when global mRNA levels are affected (such as in the FP and HDACi experiments) rather than when smaller, more specific transcript programs are affected (like in the DEX treated cells)? In other words, if there are not expected to be global changes in mRNA levels, does the hashing strategy outperform conventional methods or are the methods comparable in this situation? Could the authors please elaborate on this point?

Reviewer #4:

Remarks to the Author:

In this manuscript, Kim et al. describe an interesting approach for normalization of highly multiplexed single-cell data. They introduce a constant amount of hash oligos to each cell using a combinatorial indexing approach, effectively mimicking the use of spike-in RNA in traditional plate-based protocols. Subsequent normalization can then preserve differences in global RNA content across the cell population. The manuscript is well-written and concise, and the method itself is somewhat novel. Like spike-in normalization, its utility is probably limited, but it may still be useful on occasions involving large, biologically relevant changes in global RNA content. I have some concerns over the accuracy of the normalization, plus a few others listed below.

1. The million dollar question: are the hash oligos trapped with the same efficiency in each cell? The accuracy of the normalization depends on all cells having the same molar quantity of trapped hash oligos, especially if this is to be used to quantify changes in global RNA content between cells. It is not hard to imagine problems with systematic biases if different cell types have different permeabilities, or with inflated noise due to variation in the permeabilities within an otherwise homogeneous population. The same question has been asked of spike-ins in traditional plate-based protocols (<https://doi.org/10.1101/gr.222877.117>); a similar kind of experiment would be necessary here to demonstrate that the trapping process delivers a consistent hash oligo molarity to each cell.

2. Does the goodness of fit metric really add anything to the quality control, compared to just

filtering out cells with low total counts? It seems that cells with less sequencing depth would have lower coverage of the hash oligos, naturally leading to reduced precision and lower GoF values. Indeed, there seems to be a strong correlation between the two in Figure S1D. Figure S1E also suggests that neither metric really has a major impact in this dataset - I would be surprised if more than 10 cells were removed here.

3. I don't see what's so special about a normalized expression value of 1 in Figure 2E. A visualization of the distribution would be more informative, e.g., with violin or ridgeline plots.

4. I would suggest using a better math typesetter on page 24, as it is difficult to figure out the meaning. Some obvious questions are: why is there a log in the two mathematical expressions? Conventional size factors aren't proportional to the log-total UMIs, and I wouldn't expect the hash size factors to be proportional to the log-total hash count. What exactly is the hash duplication value, and what exactly is the RNA duplication value? What happens at a non-positive intercept? If the intercept is unfortunate enough to be zero - and I don't see why that wouldn't be possible - all factors will be undefined.

5. Where's the GEO number? I hope this omission was not intentional.

Reviewer #5:

None

Executive Summary

We are grateful to the reviewers and the editor for careful consideration and constructive criticism. Each comment is listed below in black, with our response in blue. Corresponding changes in the manuscript can be found in red. Major comments and the experiments or analyses we performed to address them are as follow:

1. Determinants of hashing oligo uptake. Our method assumes that oligo uptake is relatively consistent across cells regardless of type or treatment, but as we do not have a full understanding of the mechanism of uptake it is important to establish that this is the case. To address this concern we assessed the effects of cell size and cell cycle on hash uptake. Because we observed modest differences in hash uptake between cell cycle stages and sizes, we systematically evaluated the robustness of differential gene expression analysis in these scenarios, finding that use of our external normalization consistently outperforms conventional size factor normalization and that cell cycle and cell size have a minimal impact on DEG analysis.
2. Reproducibility of biological findings. The HDACi and HDACi/DEX co-treatment experiments were performed in duplicates, and the results were reproducible between the two replicates (see figures at the end of this document). The number of replicates were added to the figures and in the main text for clarification. The flavopiridol timecourse experiment was performed only once and we therefore validated our results by performing bulk RNA-seq (with ERCC spike-ins) in this revision. Compared with our single cell data, the repeated bulk RNA-seq experiment revealed highly similar effects of flavopiridol treatments, both at the level of global transcript abundance as well as individual gene expression.
3. Limitations and opportunities for future methods development. Although hash ladders are the first economical method for external expression quantification that works at the scale of modern single-cell RNA-seq, there remain opportunities for improvements. An expanded discussion of our method's limitations can be found in the discussion.

REVIEWER COMMENTS

Reviewer #1 (Expertise: Chromatin modification, spike-ins for scRNASeq):

In this manuscript, the authors describe the application of nuclear oligo hashing to improve quantification of single-cell RNA seq analysis. This method relies on spiking in a ladder of synthetic single strand DNA oligos to single cell nuclei preparations. These oligos get attached to the nuclei and then are co-processed in the subsequent single-cell sequencing workflow. This hashing method has previously been developed by the authors to enable barcoding for large scale single cell transcriptome studies after chemical perturbation. Here the authors use a ladder of hash oligos at different concentrations and develop computational methods to apply them for improved normalization of scRNA-seq data. The authors first show that the new normalization method improves CVs from a HEK293 scRNA-seq experiment. In a second application, they show that in contrast to normalization by conventional size factors, their method enables the detection of a global transcriptional downregulation with flavopiridol. Finally, they use HDAC inhibition in A549 cells as a model for temporal changes. They observe an initial global transcriptional downregulation, that seems compensated only later through the upregulation of acetyl-CoA biosynthetic pathways. Finally, they show that HDAC inhibition prevents the response to the glucocorticoid dexamethasone particularly at an early time point.

In summary, this manuscript provides a technical advance and novel biological data on HDAC inhibition that will be of interest to the readers of Nature Communications after the following points have been addressed:

1. Particularly for quantitative applications, it appears important to mention potential other biases that the new method might introduce. For example, uptake of the hash oligos might not be perfectly uniform across all nuclei and might correlate with other factors including nuclear size or cell cycle stage. The authors should ideally provide experimental data on single cell distribution of hash oligo uptake in the cell lines they use (e.g. using fluorescently labeled hash oligos with a FACS readout) or at least mention such potential biases in their manuscript.

We agree with the reviewer that it is important to identify and rule out potential factors that affect hash oligo uptake. In our original submission, we tested whether nuclei from distinct zebrafish cell types showed differential uptake and found minimal variation across them. Given that hash uptake is predominantly nuclear, this suggests that even cells with widely varying morphology and function have similar uptake. However, this experiment did not exclude the possibility that cells of the same type but different stages in the cell cycle might absorb hashes at different levels.

We therefore set out to assess whether cell cycle and/or cell size affect hash oligo uptake. To examine the effect of cell size, we imaged A549s labeled with hash oligos conjugated with Alexa 647. In these images, we did not observe any bias in hash intensity profile nor a clear

correlation between hash oligo intensity and cell size (Supplementary Figure 16 A-B). To correlate cell cycle and hash uptake, we used DAPI to separate cells based on cell cycle. Even though we do see a statistically significant difference in hash count distribution between cell cycle stages, we believe the difference to be modest compared to the variation in hash uptake within the cell cycle groups (Supplementary Figure 16 C). Additionally, we believe that differential expression analysis is likely robust to such modest differences. To test this assertion, we further exaggerated the difference in hash uptake by artificially increasing the total hash UMIs in A549 cells in the G2M stage up to five-fold and re-evaluated differential gene expression. When the differences in total hash UMIs between cell groups are exaggerated five-fold, the magnitude of effect size estimates and the number of DE genes that overlap with that of the original dataset are decreased by 16% (Supplementary Figure 17). Importantly, the observed difference in median total hash UMIs between G1 and G2M was ~40%, at which level the differential expression analysis were minimally impacted (53 of 57 DE genes recovered and only 5% decrease in effect size estimates). Altogether, this experiment reinforces that hash uptake is largely uniform across cell types and states with varying physical properties and that our hash-ladder based normalization is robust to any likely sources of bias based on cell cycle.

Supplementary Figure 16: Hash oligo uptake is modestly influenced by cell size and cycle stages. (A) Histogram of mean hash oligo intensity values for A549s. These cells were labeled with fluorescently labeled hash oligos and assessed via microscopy. (B) Scatterplot showing the relationship between nuclear area and mean hash oligo intensity. (C) Boxplot of \log_{10} hash UMIs for A549 and HEK293T in different cell cycle stages. These cells were labeled with DAPI and each cell cycle group was sorted by FACS. * $P < 0.05$, **** $P < 0.0001$; two-tailed t-tests.

Supplementary Figure 17: Artificial increase in hash UMIs in G2M cells does not significantly impact differential expression analysis. (A) Barplot showing the % of DE genes recovered after intentionally increasing total hash UMI counts of A549 cells in the G2M stage. (B) Boxplot comparing the ratio of effect size estimates of overlapping DE genes recovered from (A).

Nonetheless, we have revised the discussion to reiterate this core assumption of the method and allow for the possibility that it may be violated under certain circumstances.

“Importantly, the hash ladder spike-in approach assumes that uptake of hash oligos is uniform across all the nuclei in the experiment, and it is important to consider situations where this might be violated, such as with differences in cell cycle stage, cell size and/or cell or nuclear permeability. While it is challenging to correlate permeability and hash uptake, we believe that the effect of cell cycle/size on hash ladder normalization is minimal and found that an observed difference in hash uptake did not significantly alter differential expression analysis between these cells (**Supplementary Fig. 16 and 17**).”

2. For the HDAC inhibitor treatment, the authors describe a 60% drop in overall mRNA levels in their early 1 and 3 hour timepoints. How would this effect be compatible with average mRNA half-lives in mammalian cells? It is important to validate this effect in an independent method, either on bulk cells or using a nascent transcription method.

We think the reviewer raises a valuable point regarding the observed early drop (albeit closer to a 40% reduction) in recovered nuclear mRNA upon HDACi treatment. Importantly, this result was observed when ordering cells over a continuous “pseudotime” trajectory (Supplementary Figure 7A). Indeed, when grouping cells by discrete treatment times, either using the same single cell RNA-seq data (Supplementary Figure 7B), or with newly generated bulk RNA-seq (Supplementary Figure 7C), this reduction was undetectable. This agrees with our previous observation that cellular response to HDACi is vastly heterogeneous and suggests that cell-to-cell variability within each treatment might mask these state-specific effects. We believe that validating this result via an orthogonal assay is challenging, considering that it requires absolute quantification of global RNA levels.

Given that our method measures primarily nuclear, poly-adenylated RNAs, we suspect that the observed reduction in recovered mRNA upon HDACi treatment likely represents an inhibition of nascent transcription, consistent with previous observations (Greer et al., Molecular Cell, 2013). Notably, our flavopiridol treatment (which blocks transcription elongation) also indicates >40% reductions in nuclear mRNA within a similar time frame (Fig. 2B). However, we cannot rule out the alternative possibility that decay or cytoplasmic export of mRNA is enhanced by our HDACi treatments.

Supplementary Figure 7: Transient reduction of total RNA UMIs along the HDACi trajectory is only visible in the pseudotime space. (A) Scatterplot of total RNA UMIs as a function of inferred pseudotime values. (B) Boxplot showing the distribution of total RNA UMIs of cells grouped by HDACi treatment time. (C) Barplot of total RNA UMIs measured by bulk RNA-seq. Colors indicate HDACi treatment time.

3. For the dexamethasone treatment, the UMAP representations in Fig. 4B might be misleading as the non-HDAC inhibitor treatment condition is shown separately whereas the two time points of HDAC inhibitor treatment are shown together, whereby treatment time is clearly the main driver of separation. Either all experiments (no HDAC inhibitor, 4 h, 24 h treatment) should be shown in one plot, or separate plots should be shown for all doses/timepoints always comparing vehicle vs. dexamethasone as done for the non-HDAC inhibitor treated condition. The chosen HDAC inhibitor concentrations appear too high to elicit a differential effect.

We agree with the reviewer. We changed Fig. 4B so that the UMAPs from all the experiments are shown. Additionally, we added UMAP plots that show all doses/timepoints in the Supplemental Information.

Fig. 4B: UMAP embeddings of HDACi and DEX treated cells, showing all the cells in one plot.

Supplementary Figure 11: UMAP embeddings of HDACi and DEX treated cells. UMAP embeddings from Figure 4B were faceted by HDACi dose and treatment time to compare vehicle and DEX treated cells.

4. From Fig. 4F it appears that at least some of the genes that are unresponsive to dexamethasone treatment in the presence of HDAC inhibitors are unresponsive because HDAC inhibition already causes a similar effect to dexamethasone treatment. Eg. DOCK4 appears upregulated by HDAC inhibition alone already, and no further upregulation occurs with dexamethasone. Such effects cannot be recognized from the representation chosen in Fig. 4D, and it would be desirable to include an additional heatmap.

Indeed there are a handful of unresponsive DEX genes as a consequence to the previous HDACi treatment. We have now attempted to more clearly define how these two drug treatments affect these “unresponsive” genes and included them in new comments in the main text, as well as a Supplemental Figure 15 (shown below). Briefly, we categorized these genes into three classes to reflect how HDACi influences their expression levels: saturated, dominated, or attenuated. Saturated indicates the gene is differentially expressed as a consequence of HDAC inhibition and in the same direction as the response to DEX alone. Similarly, a gene was defined as dominated (meaning expression changes were dominated by the HDAC inhibitor) if the gene is differentially expressed as a consequence of HDAC inhibition but in the opposite direction as DEX alone. Finally, an unresponsive DEX gene was classified as attenuated if the expression is not significantly altered by HDAC inhibition alone, but nonetheless incapable of responding to DEX after HDACi.

“Interestingly, we observed various classes of effects on these unresponsive genes suggesting multiple ways in which HDACi might interfere with the DEX response (Supplementary Fig. 15).”

Supplementary Figure 15: Classification of normally DEX-induced genes that failed to respond to DEX as a consequence of HDAC inhibition. (A) Illustration of gene expression patterns in each gene category. (B) Proportion of unresponsive DEX genes for each category, colored by HDAC inhibitor treatment time. (C) Heatmap of beta coefficient values of

unresponsive DEX genes after 4 hours (left) and 24 hours (right) of HDACi treatment. Rows represent genes and columns represent beta coefficients (FDR < 0.01) for terms that capture effects from HDACi treatment alone, DEX treatment alone and DEX treatment after HDACi treatment.

5. I am not fully clear how the treatment regimen for HDAC inhibitors and dexamethasone was performed. I think it was 4 hours HDAC inhibitor treatment followed by 2 hours of HDAC inhibitor plus dexamethasone, rather than two plus two hours. In this case the relevant reference is 6 h HDAC inhibitor treatment, which may be too long given that the authors already detect compensatory mechanisms at this time point.

We apologize for the lack of clarity. For the HDACi/DEX experiment, we added DEX 2 hours prior to the end of the HDACi treatment. For example, the 4 hour timepoint was performed by 2 hour HDACi treatment followed by 2 hours of DEX treatment. Given the results in the HDACi timecourse experiment, 4 hour HDACi treatment will not completely induce acetate starvation related compensatory effects. We revised the text so that the explanation of the method is clear.

“Similarly, for the HDAC inhibitor and dexamethasone co-treatment experiment, we added 1 μ L of 100 μ M or 1 μ L of 1 mM of either abexinostat or pracinostat to each 96 well to attain a final concentration of 1 and 10 μ M, respectively. Dexamethasone (Selleck Chemicals) was added at a concentration of 100 nM (1 μ L of 10 μ M) two hours before the end of the HDAC inhibitor treatment.”

Reviewer #2 (Expertise: Principles of transcriptional regulation, scRNASeq):

The manuscript describes the potential to utilize nuclear hash oligo as a 'spike-in' control for RNA quantification from single-nuclei RNA-seq. By deriving the size factor from the fitted slope and intercept of hashing oligo ladders, and applying them to the situation where the global transcriptional changes are expected, authors have shown nicely that nuclear hash oligo based normalization results in increased power for the detection of differentially expressed genes.

This manuscript address very important topic in single-cell analysis and elegantly conducted overall. I have few minor comments to improve the manuscript for the publication.

1. Justifying the logics for the size factor calculation.

Could the authors test their size factor calculations, by testing the inclusion/exclusion of each factors and showing whether it changes the conclusion (in terms of differential gene expression testing)?

We clarified our reasoning for size factor normalization in the Method section (see below). Briefly, we added cells' RNA and hash duplication rates in order to adjust for sequencing depth within and between scRNA-seq libraries. Total hash UMIs are included instead of total RNA UMIs to account for technical bias introduced in the library. Similarly, we interpreted the slope and intercept of the calibration curves as library preparation efficiency and a fraction of library used for sequencing, respectively. Cells that are low-quality (indicated by low slope and more negative intercept values) have smaller size factor values compared to high-quality cells and their gene expression values are scaled accordingly.

To address the reviewers comment, we tested the exclusion of each parameter in our size factor calculation using the flavopiridol data as a reference. For this analysis, size factors were calculated while individually omitting the slope or intercept values derived from the hash ladder calibration lines, or the total hash counts. With these simplified size factors for each cell, we repeated the differential expression analysis across the FP treatment time course. Ultimately, excluding these parameters from size factor calculations yielded a similar number of DE genes and recovered a majority of genes that are downregulated by FP, suggesting that our size factor calculation is robust to the variation in each of these parameters (see figure below). We believe that the parameters mentioned above provide additional accountability for technical variation that can occur in sci-RNA-seq experiments (Kim et al., Nature Communications, 2015). For example, systematic errors in pooling during split-and-pool barcoding would alter the intercept, while an inefficient reverse transcription reaction in a subset of reaction wells might affect the slope of recovered hash calibration curves.

“With the assumption that the nuclear capture rate of the hash oligos are approximately equal across cells, we reasoned that technical variation should mostly account for the differences in the total number of observed hash UMIs. The distribution of total hash UMIs were log-normally distributed and therefore logarithm of total hash UMIs used for the calculation. Additionally, we

reasoned that the slope and intercept values of the hash ladder calibration curves reflect the cells' library preparation efficiency and the amount of input RNA/hash ladder molecules used for the library and sequencing¹⁸, respectively. For example, variations in reverse transcription (RT) reaction efficiency will influence the capture of lowly expressed RNA molecules (slope) and systematic errors in pipetting and pooling steps will affect the number of molecules captured in each reaction (intercept). The duplication rates of the hash oligos and transcriptomes were included in the size factor calculation to correct for differences in sequencing depth between samples and multiple sequencing runs. As a result, nuclei that exhibit high technical variation as indicated by the hash ladder parameters would have low size-factor values to compensate for low library preparation efficiency.”

The effect of omitting parameters in the size factor calculation on flavopiridol timecourse data. (A) Barplots showing the number of DE genes obtained using different size factor values. We tested our hash ladder size factor calculations by omitting total hash UMIs (“Total hash”), intercept (“Intercept”), slope (“Slope”), or removing both slope and intercept (“Just hash”). We normalized the gene expression data using the respective size factor values and performed differential gene expression analysis. (B) The percent recovery of DE genes found in (A), using the “full” hash ladder size factor calculation as a reference.

2. Extrapolating the size factor normalization to the case where you don't have spike-in
 In most experiments, there are still no spike-in included. Could authors investigate whether the true size factor calculated based on spike-in can be derived/explained from the values that can be obtained in more general single-cell RNA-seq techniques (such as total RNA UMIs, number of genes detected)? This can be done by simply showing correlation plots or fitting regression models. If the authors could suggest some general rule that can improve the current 'crude' size factor calculation, that it can have a greater impact on the field.

We agree that extrapolating our discovered normalization values for these experiments to others is a highly attractive idea. In fact, we previously have shown that relative transcript counts can be computationally inferred from relative expression values without the use of spike-ins in scRNA-seq data (Qiu et al., Nature Methods, 2017). However, it is still highly desirable to have spike-ins, as computational approaches cannot control for biases introduced during the sample preparation steps.

Additionally, we think extrapolations are biased for experiments on unseen conditions where we cannot anticipate true differences in transcriptome content. For example, it would be difficult (without prior knowledge) to identify the global reduction in transcript abundance upon FP treatment. While the FP response is somewhat of an extreme example, we think it makes a good case for the use of measurable external controls.

3. Could the authors discuss about the usability of this hashing methods? i.e. could this be applied in other platform including 10X Genomics, and what will be the general success rate of adding right concentration of spike-in. (as having it too much could potentially ruin your experiment)

We have now added comments on the cross platform compatibility and limitations of the hashing method to our Discussion. Currently, our nuclear hashing strategy relies on PFA fixation, which poses problems for the standard 10x Genomics platform. However, it was recently reported that transcriptomes of formaldehyde-fixed cells could be profiled with the 10x Genomics platform, using non-standard processing, providing an avenue for using hashing with such tools (scfi-RNA-seq; Datlinger et al., Nature Methods, 2021). We have yet to implement hashing on other scRNA-seq platforms, but we are also optimistic that hashing will be compatible with plate-based scRNA-seq methods.

As suggested by the reviewer, recovering enough hash reads per cell in these experiments is crucial in order to properly assign cells to their experimental conditions and determine ladder-based size factors. However, because of the small size of amplified hash molecules (200 bp) compared with transcriptome fragments (avg > 300bp), it is possible to separate these library components prior to sequencing. While we previously determined the optimal range of hash concentration for a typical scRNA-seq experiment (Srivatsan et. al. Science 2020), adding “too much” hash can usually be compensated for by size selection and separate purification of hash and cDNA libraries prior to sequencing.

“Furthermore, hash oligos can be captured using aldehyde and alcohol fixatives (MeOH; unpublished result); therefore, hashing should be compatible with other scRNA-seq platforms.”

4. (optional) cell cycle systems (such as the one with FUCCI reporter) could be perfect one to show the power of this approach

We agree with this suggestion and have attempted to address the point in our response to Reviewer #1 P1, where we mainly assessed how cell cycle affects hash oligo uptake (see figure below). In this experiment, we used DAPI instead of the FUCCI reporters to sort for cells in different cell cycle stages (G1, S, and G2M). In both HEK293Ts and A549s, we observed a modest difference in hash count distribution and global RNA level differences between cell cycle phases. Interestingly, when comparing gene expression changes between G1 and G2M, a higher number of upregulated genes were recovered with hash ladder normalization. This result is consistent with an increase in mRNA content during cell cycle progression (Vallejos et al., *Genome Biology*, 2016) and demonstrates the power of using hash ladder normalization for detecting global expression changes.

Single cell transcriptome analysis of cell cycle stages using the hash ladder. (A) Flow cytometry plot showing gating strategy used to sort for cell cycle stages. (B) Boxplot of \log_{10} hash UMIs for A549 and HEK293T in different cell cycle stages. These cells were labeled with DAPI and each cell cycle group was sorted by FACS. * $P < 0.05$, **** $P < 0.0001$; two-tailed t-tests. (C) Barplot showing the number of differentially expressed genes in cell cycle stages using G1 as a baseline.

Reviewer #3 (Expertise: Transcriptional regulation at the genome level):

Kim et al., present a novel spike-in “hashing” strategy that normalizes scRNA-seq data to correct for technical variations without having to assume that total mRNA levels are similar between individual cells, an important point when comparing conditions that affect global transcript levels. The authors demonstrate that the hashing strategy detects global transcript changes (i.e., after treatment with inhibitors), while conventional normalization methods fail to capture these global changes. However, the hashing and conventional strategies produce comparative results when small, more specific transcript changes occur (Figure 4 and Supp Fig.9).

Given the lack of spike-in normalization methods currently available for scRNA-seq data and the gain in popularity of scRNA-seq experiments, this study is timely and important for the field. The study is well designed and presented in its current form, but the manuscript would benefit from clarification of a few points and minor additions to the analysis prior to publication.

1. In Supplementary Figure 3D only ~25-28 unique hash oligos were recovered per cell out of the 48 used in the hash ladder. Is there a bias in the oligos that are not recovered across cells or do different oligos fail to sequence across different cells? If there is a bias in the oligos that are not recovered, then is it necessary to include them in the ladder? If random oligos fail to sequence across individual cells, do differences in which oligos are recovered affect normalization?

We included a large range of concentrations in our ladder (3 orders of magnitude) in order to capture the dynamic range of our single cell transcript measurements. As with the eight-molecule ladder, we observe a strong correlation between starting molecule abundance and final recovery, with lower abundance molecules more often absent within individual cells. Because we know the concentration of all hashes with the ladder, the drop-out of our lower-abundance molecules can serve as a proxy for concentrations at which endogenous transcripts are likely to be lost from the assay as well. In fact, several groups have used the drop-out of low abundance spike-in molecules in their noise models (Grun et al., *Nature Methods*, 2014; Kim et al., *Nature Communications*, 2015). Importantly, we have also previously determined that differences in the 10bp hash ID sequence do not bias nuclear uptake (Srivatsan et al., *Science*, 2020).

2. In Figure 3C, 446 genes were used for the hierarchical clustering that revealed four kinetically distinct groups enriched in various gene ontologies, yet in Supp Figure 6A there are 1,485 differentially expressed genes identified by hash normalization. What is the rationale for only using 446 genes for the clustering and gene ontology analysis in Figure 3C?

The DE genes used in Supplemental Figure 8A (previously Supplementary Figure 6) include all DE genes with a FDR of $1e-2$, which is the cutoff we used for other experiments. For

hierarchical clustering, we used a more stringent FDR of 1e-10 to hone in on the genes that are more dramatically affected by HDACi.

3. Which ontology terms are enriched for the 643 genes that were identified by both the conventional and hash normalization methods after HDACi treatment (Supp Figure 6A)? Do most of the genes involved in acetyl-CoA biosynthesis fall into the 643 genes commonly identified by the two methods or are they mostly found in the gene set only identified by the hashing strategy?

Both conventional and hash ladder based normalization strategies identified acetate metabolism, immune response and cell cycle as GO terms affected by HDACi treatments. However, as shown in Supplemental Figure 8C (previously Supplementary Figure 6), more of the genes involved in acetyl-CoA biosynthesis and previously found to be affected by HDACi (Srivatsan et al., Science, 2020) are identified by the hash ladder normalization method.

4. Could the authors please elaborate on the following statement and figure? “Additionally, the UMAP embedding of gene expression profiles using the hash ladder-based normalization approach displayed a more pronounced branching of cells undergoing cell cycle related changes as a result of HDAC inhibition, which is less well defined in the embedding obtained with conventional normalization (Fig. 3A and Supplementary Fig.7).”
What is the significance of a more pronounced branching of cells? This point may not be very clear for readers that are not as familiar with this type of analysis.

HDAC inhibitors induce cell cycle effects (Kim et al., The Journal of Antibiotics, 2000; Finzer et al., Oncogene, 2001), and we therefore expected to see a group of cells that exhibit cell cycle arrest in the UMAP embedding. In Fig. 3A, we wanted to highlight that hash ladder normalization accentuates HDACi-related cell cycle effects in the embedding compared to conventional normalization. However, we reviewed our analysis and came to a conclusion that comparing UMAP visualizations can be subjective. We therefore removed these sentences from the main text.

“Moreover, in line with established perturbations of proliferation by HDACi^{40,41}, we observed a group of HDACi-treated cells with altered gene expression relating to cell cycle effects (Supplementary Fig. 9).”

5. The majority of detected differentially expressed DEX genes are identified by both conventional and hash normalization, with ~20 genes that show opposite responses (Supp Fig 9A and B). Could the authors please clarify if the majority of expected DEX response genes were detected using both normalization methods or are most DEX responsive genes present in the ~20 genes that showed opposite responses between the two normalization methods?

A set of DEX response genes from Reddy et al., Genome Research, 2009 were used for differential gene expression testing, and a majority of them were detected using both normalization methods as shown in Supplementary Figure 12A and B (previously Supplementary Figure 9). We detected fewer upregulated genes but more down-regulated genes using hash ladder normalization, but no genes showed opposite responses between the normalization methods. Thus, the ~20 genes alluded to are not DEX genes that show opposite responses. Rather, these genes are uniquely identified by each normalization approach.

6. Do the authors have any insight into why ~20 genes displayed opposite responses between the two normalization methods after DEX treatment (Supp Fig 9A and B)? Is there something about the expression of these genes that make them more susceptible to differences between the normalization methods?

Like mentioned in #5, these genes are uniquely found using the respective normalization methods, not genes that show opposite responses. We investigated these genes and found that they do not share a common biological function. In our analysis, we recovered a higher number of genes that are downregulated due to DEX treatment using the hash ladder normalization than that of conventional normalization (Supplementary Figure 12B).

7. If the majority of expected DEX response genes are identified by both normalization methods, does this suggest that the main benefit of the hash normalization method is to normalize transcript counts when global mRNA levels are affected (such as in the FP and HDACi experiments) rather than when smaller, more specific transcript programs are affected (like in the DEX treated cells)? In other words, if there are not expected to be global changes in mRNA levels, does the hashing strategy outperform conventional methods or are the methods comparable in this situation? Could the authors please elaborate on this point?

This is an important point. Indeed, our data suggests that in cases of subtle or gene specific changes in transcription, it appears that conventional normalization does perform quite well and that the additional benefits of using our spike in approach may only be marginal. However, because it is impossible to predict the transcriptional outcome of previously unseen conditions (particularly in the case of gene knockdowns or knockouts), using an external spike-in standard will guard against any assumptions made with conventional normalization, whether valid or not. We have updated the text to better reflect these points.

“Our analysis suggests that hash ladder based normalization is particularly beneficial when changes in global transcriptional levels are expected (e.g. flavopiridol). By contrast, when only a small number of genes were affected by a treatment, conventional normalization performed similarly to the hash ladder based normalization (e.g HDACi and HDACi/DEX). However, because it is often difficult or impossible to predict the outcome of previously unseen perturbations, the hash ladder provides a robust normalization approach, unbiased by a priori assumptions.”

Reviewer #4 (Expertise: scRNASeq analysis with Spike-ins):

In this manuscript, Kim et al. describe an interesting approach for normalization of highly multiplexed single-cell data. They introduce a constant amount of hash oligos to each cell using a combinatorial indexing approach, effectively mimicking the use of spike-in RNA in traditional plate-based protocols. Subsequent normalization can then preserve differences in global RNA content across the cell population. The manuscript is well-written and concise, and the method itself is somewhat novel. Like spike-in normalization, its utility is probably limited, but it may still be useful on occasions involving large, biologically relevant changes in global RNA content. I have some concerns over the accuracy of the normalization, plus a few others listed below.

1. The million dollar question: are the hash oligos trapped with the same efficiency in each cell? The accuracy of the normalization depends on all cells having the same molar quantity of trapped hash oligos, especially if this is to be used to quantify changes in global RNA content between cells. It is not hard to imagine problems with systematic biases if different cell types have different permeabilities, or with inflated noise due to variation in the permeabilities within an otherwise homogeneous population. The same question has been asked of spike-ins in traditional plate-based protocols (<https://doi.org/10.1101/gr.222877.117>); a similar kind of experiment would be necessary here to demonstrate that the trapping process delivers a consistent hash oligo molarity to each cell.

We agree that uniformly capturing hash oligos across cells is important for our method and to ensure accurate global RNA content estimation. We know from previous imaging data in our lab that hashes go mainly to the nucleus, even in permeabilized cells (Srivatsan et al., Science, 2020). In this work, we showed that nuclei from very different cells, such as various cell-types from dissociated zebrafish embryos have similar uptake (Supplementary Figure 2). In our response to reviewers 1 and 2, we have now included results showing that cell cycle and cell size at most modestly impact hash uptake (Supplementary Figure 16, reproduced below).

While investigating the relationship between variation in cell/nuclear permeability and hash uptake would be intriguing, we think this property would be challenging to control in the lab, but we now acknowledge this as a potential confounder in the Discussion. Importantly, while perhaps limited, we believe our strategy is unique in its ability to easily scale to thousands of profiled cells. Ultimately, in experimental systems where differential permeability might be a concern, conventional normalization is always available and hash ladders cost very little.

“Importantly, the hash ladder spike-in approach assumes that uptake of hash oligos is uniform across all nuclei in the experiment, and it is important to consider situations where this might be violated, such as with differences in cell cycle stage, cell size and/or cell or nuclear permeability. While it is challenging to correlate permeability and hash uptake, we believe that the effect of cell cycle/size on hash ladder normalization is minimal and found that an observed difference in

hash uptake did not significantly alter differential expression analysis between these cells (Supplementary Fig. 16 and 17).”

Supplementary Figure 16: Hash oligo uptake is modestly influenced by cell size and cycle stages. (A) Histogram of mean hash oligo intensity values for A549s. These cells were labeled with fluorescently labeled hash oligos and assessed via microscopy. (B) Scatterplot showing the relationship between nuclear area and mean hash oligo intensity. (C) Boxplot of \log_{10} hash UMIs for A549 and HEK293T in different cell cycle stages. These cells were labeled with DAPI and each cell cycle group was sorted by FACS. * $P < 0.05$, **** $P < 0.0001$; two-tailed t-tests.

2. Does the goodness of fit metric really add anything to the quality control, compared to just filtering out cells with low total counts? It seems that cells with less sequencing depth would have lower coverage of the hash oligos, naturally leading to reduced precision and lower GoF values. Indeed, there seems to be a strong correlation between the two in Figure S1D. Figure S1E also suggests that neither metric really has a major impact in this dataset - I would be surprised if more than 10 cells were removed here.

We have observed that cells with less sequencing depth have a higher likelihood of having lower GoF values and these cells are naturally removed after filtering for total hash UMIs. However, we believe that the goodness of fit adds an extra layer of quality control to filter out low-quality cells. The cells should have received hash oligos proportional to their abundances and the recovery of hash UMIs that deviate from the initial concentrations suggest poor sample preparation. Therefore, we believe they should be omitted for hash size factor calculations and downstream analysis.

It is true that only 9 out of 1888 cells are removed in the dataset shown in Supplementary Figure 1 and none of the cells are removed for the flavopiridol dataset. However, we were able to filter out 725 and 1359 cells in the HDACi and HDACi/Dex datasets, respectively, before further removing cells with low total RNA counts. Similar to the results in Supplementary Figure 1D, the cells with lower GoF values in these datasets tended to have lower numbers of total RNA and hash UMIs.

3. I don't see what's so special about a normalized expression value of 1 in Figure 2E. A visualization of the distribution would be more informative, e.g., with violin or ridgeline plots.

We agree that showing distributions of expression levels will be more helpful rather than normalized expression value > 1. However, we chose the latter because the single-cell transcriptome data are very sparse and a majority of the cells have expression value of 0 (plot A). As shown below, the violin plots for the genes in Fig. 2E are concentrated at a value of 0 rather than focusing on the populations of cells with high expression (plot B). Plotting the expression data after filtering out cells with zero expression values is misleading because only a small number of cells have expression > 0 (plot C).

Visualization of CD151 and LAMC1 expression values. CD151 and LAMC1 expression values were plotted by taking the percentage of cells with normalized expression > 1 (A) and using violin plots without (B) and with (C) filtering out cells with zero expression values.

4. I would suggest using a better math typesetter on page 24, as it is difficult to figure out the meaning. Some obvious questions are: why is there a log in the two mathematical expressions? Conventional size factors aren't proportional to the log-total UMIs, and I wouldn't expect the hash size factors to be proportional to the log-total hash count. What exactly is the hash duplication value, and what exactly is the RNA duplication value? What happens at a non-positive intercept? If the intercept is unfortunate enough to be zero - and I don't see why that wouldn't be possible - all factors will be undefined.

We have now switched to the LaTeX math typesetter to make the equations more readable (see below). Also, the reviewer is correct in that conventional size factors are proportional to total UMIs, not $\log(\text{total UMIs})$. This error is corrected in the revised manuscript.

We observed that the distribution of total hash UMIs is log-normal and therefore reasoned that it was more appropriate to use $\log(\text{total hash UMIs})$ for the size factor calculation (see below figure A,B). When $\log(\text{total hash UMIs})$ was used, UMAP embedding showed a more clear flavopiridol treatment time dependent structure (figure C) than without the log-transformation. Differential expression analysis of the flavopiridol time series data revealed a higher number of downregulated genes in comparison to the conventional normalization, regardless of whether logarithm of total hash UMIs was taken (figure D).

The duplication rates indicate how saturated a library is after sequencing (when you start sequencing the same molecules over and over) and they are used to assess the sequencing depth for each cell. We include them in our size-factor calculation to adjust for the differences in sequencing depth of the cells within and between sci-RNA-seq libraries.

For our analysis, none of the cells had non-positive intercept values. We believe that the cells in this category are rare and we speculate that these cells could arise as a result of variation in capture rate or purely due to a fitting error.

We have updated the manuscript to include a deeper description of the intuition behind these various design choices in the hadder ladder normalization method, reproduced below.

“With the assumption that the nuclear capture rate of the hash oligos are approximately equal across cells, we reasoned that technical variation should mostly account for the differences in the total number of observed hash UMIs. The distribution of total hash UMIs were log-normally distributed and therefore logarithm of total hash UMIs used for the calculation. Additionally, we reasoned that the slope and intercept values of the hash ladder calibration curves reflect the cells’ library preparation efficiency and the amount of input RNA/hash ladder molecules used for the library and sequencing¹⁸, respectively. For example, variations in reverse transcription (RT) reaction efficiency will influence the capture of lowly expressed RNA molecules (slope) and systematic errors in pipetting and pooling steps will affect the number of molecules captured in each reaction (intercept). The duplication rates of the hash oligos and transcriptomes were included in the size factor calculation to correct for differences in sequencing depth between samples and multiple sequencing runs. As a result, nuclei that exhibit high technical variation as indicated by the hash ladder parameters would have low size-factor values to compensate for low library preparation efficiency.”

Justification of using log(hash UMIs) in the size factor calculation. (A) Distribution of total hash UMIs in the flavopiridol data. (B) Distribution of size factor values without (left) and with (right) taking the log of total hash UMIs in the size factor calculation. (C) UMAP embeddings of flavopiridol timecourse data using the size factor values using the non-logged (top) and logged (bottom) total hash UMIs. (D) Barplot comparing the number of flavopiridol-related DE genes identified using different size factor values.

5. Where's the GEO number? I hope this omission was not intentional.

The data were not deposited to GEO before the initial submission. Manuscript is updated so that it includes the GEO number (GSE166470).

Supplementary Figure 5: Bulk RNA-seq corroborates scRNA measurements in flavopiridol timecourse experiment. (A) Log₁₀ total RNA UMIs for cells treated with flavopiridol at each timepoint, measured by bulk RNA-seq. $n = 2$ replicates. (B) Correlation of expected and observed ERCC spike-in molecules for vehicle cells. (C) Number of differentially expressed genes found using conventional and ERCC normalization. (D) Scatterplots of statistically significant gene estimates derived from sci-RNA-seq data using monocle3 and estimates derived from the bulk RNA-seq data using DESeq2. Black line indicates $y = x$ and the blue line indicates the fit.

Technical reproducibility of the HDACi timecourse experiment. Technical replicates were treated separately and Figure 3 is reproduced. (A) UMAP embedding of HDACi treated cells. (B) Scatterplot showing total RNA UMIs as a function of pseudotime showing a transient reduction in total RNA UMIs. (C) Pseudotime distribution of HDACi treated cells, red line indicating the onset of acetyl-CoA deprivation. (D) Hash ladder normalized expression values of ACSL3 and SLC2A3.

Technical reproducibility of the HDACi and DEX co-treatment experiment. Technical replicates (n = 2) were treated separately and Figure 4 is reproduced. (A) UMAP embedding of HDACi and DEX treated cells. (B) Scatterplot showing log₂ fold changes of DEX responsive genes (DEX / vehicle) without and with preceding HDAC inhibitor treatment. (C) Scatterplot showing the correlation of log fold change ratio (DEX / HDACi + DEX) from (B) for the replicates 1 and 2. Black dotted line indicates y=x and the red line indicates the fit. (D) Hash ladder normalized expression values of DEX genes (*DOCK4* and *UGDH*) that do not respond in cells have received the HDAC inhibitors.

Reviewers' Comments:

Reviewer #1:

Remarks to the Author:

In this revised version, the authors have addressed my comments, added additional experimental data and a discussion of possible limitations and best applications of the new method. I now support publication.

Reviewer #2:

Remarks to the Author:

In this revised manuscript, all the raised points are well answered/tried, including the cell cycle experiment which adds additional value to the study. I recommend this manuscript for publication in Nature communications, with some minor suggestions as following:

With regards to point 3, which is about general usability, it's not accurate to describe that "hashing should be compatible with other scRNA-seq platforms", as the fixation is not a universal/well-confirmed option for many platforms yet. I would suggest to modify this description.

Reviewer #3:

Remarks to the Author:

All of my previous concerns have been addressed in this revision and no further revisions are necessary for publication.

Reviewer #4:

Remarks to the Author:

I'm pretty satisfied with the revisions and I don't have anything more to say.

We are grateful to the reviewers and the editor for careful consideration and constructive criticism. Each comment is listed below in black, with our response in blue. Corresponding changes in the manuscript can be found in red.

Reviewer #1 (Remarks to the Author):

In this revised version, the authors have addressed my comments, added additional experimental data and a discussion of possible limitations and best applications of the new method. I now support publication.

Reviewer #2 (Remarks to the Author):

In this revised manuscript, all the raised points are well answered/tried, including the cell cycle experiment which adds additional value to the study. I recommend this manuscript for publication in Nature communications, with some minor suggestions as following:

With regards to point 3, which is about general usability, it's not accurate to describe that "hashing should be compatible with other scRNA-seq platforms", as the fixation is not a universal/well-confirmed option for many platforms yet. I would suggest to modify this description.

As suggested by the reviewer, we revised the text in Discussion to specify that hashing is only available on scRNA-seq platforms that are compatible with fixation:

"Furthermore, hash oligos can be captured using aldehyde and alcohol fixatives (MeOH; unpublished result); therefore, hashing could be implemented in other scRNA-seq platforms that are compatible with chemical fixation."

Reviewer #3 (Remarks to the Author):

All of my previous concerns have been addressed in this revision and no further revisions are necessary for publication.

Reviewer #4 (Remarks to the Author):

I'm pretty satisfied with the revisions and I don't have anything more to say.